# IndustryEQA: Pushing the Frontiers of Embodied Question Answering in Industrial Scenarios

Yifan Li[1]*, Yuhang Chen[2]*, Anh Dao[1]*, Lichi Li[3], Zhongyi Cai[1],
Zhen Tan[4], Tianlong Chen[2], Yu Kong[1]

[1]Michigan State University, {liyifa11, anhdao, caizhon2, yukong}@msu.edu
[2]University of North Carolina at Chapel Hill, {yuhang, tianlong}@cs.unc.edu
[3]Independent researcher, lichili233@gmail.com
[4]Arizona State University, ztan36@asu.edu

## Abstract

Existing Embodied Question Answering (EQA) benchmarks primarily focus on household environments, often overlooking safety-critical aspects and reasoning processes pertinent to industrial settings. This drawback limits the evaluation of agent readiness for real-world industrial applications. To bridge this, we introduce IndustryEQA, the *first* benchmark dedicated to evaluating embodied agent capabilities within safety-critical warehouse scenarios. Built upon the NVIDIA Isaac Sim platform, IndustryEQA provides high-fidelity episodic memory videos featuring diverse industrial assets, dynamic human agents, and carefully designed hazardous situations inspired by real-world safety guidelines. The benchmark includes rich annotations covering six categories: equipment safety, human safety, object recognition, attribute recognition, temporal understanding, and spatial understanding. Besides, it also provides extra reasoning evaluation based on these categories. Specifically, it comprises 971 question-answer pairs generated from small warehouse and 373 pairs from large ones, incorporating scenarios with and without human. We further propose a comprehensive evaluation framework, including various baseline models, to assess their general perception and reasoning abilities in industrial environments. IndustryEQA aims to steer EQA research towards developing more robust, safety-aware, and practically applicable embodied agents for complex industrial environments. The project is available[2].

## 1 Introduction

Embodied Artificial Intelligence (AI) aims to develop agents that perceive, reason, and interact intelligently within the physical world. A key aspect is understanding from an egocentric view by interpreting open-ended language grounded in common-sense knowledge. Evaluating such complex environmental understanding continues to pose a substantial challenge.

Embodied Question Answering (EQA) [7] has emerged as a comprehensive task for this evaluation, requiring an agent to explore its surrounding environment, collect visual evidence, and answer natural language questions. This task encompasses multiple essential capabilities, including episodic memory, purposeful navigation, visual perception, spatial reasoning, *etc*. Recently, studies such as OpenEQA [23] have begun investigating the open-vocabulary EQA problem within more realistic settings.

---

*Equal contribution

[2]https://jackyfl.github.io/IndustryEQA_project_page/

39th Conference on Neural Information Processing Systems (NeurIPS 2025) Track on Datasets and Benchmarks.

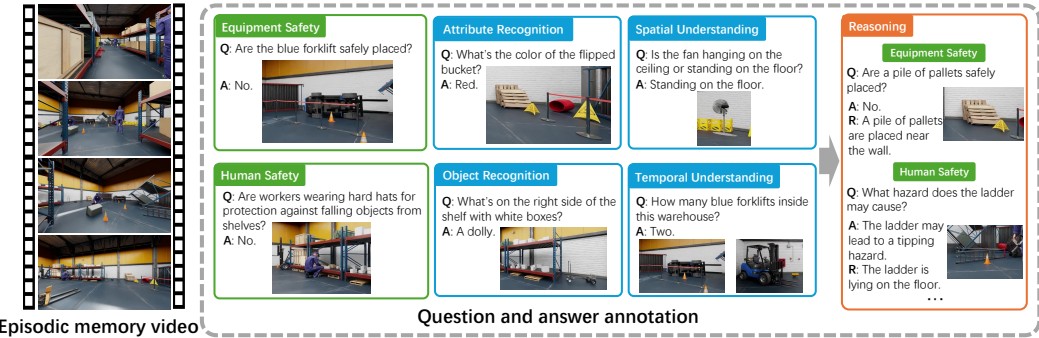

Figure 1: An illustration of the IndustryEQA benchmark, consisting of episodic memory videos and annotations. IndustryEQA annotations incorporate six types of annotations, covering safety (equipment safety and human safety) and general perception capabilities (object recognition, attribute recognition, temporal understanding and spatial understanding). Furthermore, it also incorporates extra reasoning answers for the questions that require deeper thinking.

Table 1: Comparison with current EQA benchmarks. "Human" indicates whether the benchmark includes human subjects, and "EM" stands for episodic memory.

| Benchmark | Scenario | Platform | Safety | Reasoning | Open Vocab | Human | LLM scoring | EM Video |
|---|---|---|---|---|---|---|---|---|
| EQA-v1$_{CVPR18}$ [7] | Household | House3D | ✗ | ✗ | ✗ | ✗ | ✗ | ✗ |
| IQUAD$_{CVPR18}$ [13] | Household | AI2-THOR | ✗ | ✗ | ✗ | ✗ | ✗ | ✗ |
| MP3D-EQA$_{CVPR19}$ [44] | Household | Matterport3D | ✗ | ✗ | ✗ | ✗ | ✗ | ✗ |
| MT-EQA$_{CVPR19}$ [49] | Household | House3D | ✗ | ✗ | ✗ | ✗ | ✗ | ✗ |
| ScanQA$_{CVPR22}$ [2] | Household | RGB-D Camera | ✗ | ✗ | ✗ | ✗ | ✗ | ✓ |
| SQA3D$_{ICLR23}$ [22] | Household | ScanNet | ✗ | ✗ | ✗ | ✗ | ✗ | ✗ |
| K-EQA$_{TPAMI23}$ [39] | Household | AI2-THOR | ✗ | ✗ | ✓ | ✗ | ✗ | ✗ |
| Robo-VQA$_{ICRA24}$ [36] | Household | RGB Camera | ✗ | ✗ | ✗ | ✗ | ✗ | ✓ |
| OpenEQA$_{CVPR24}$ [23] | Household | ScanNet/HM3D | ✗ | ✗ | ✓ | ✗ | ✓ | ✓ |
| CityEQA-EC$_{ArXiv25}$ [50] | City | EmbodiedCity | ✗ | ✓ | ✓ | ✗ | ✓ | ✗ |
| IndustryEQA | Warehouse | Isaac Sim | ✓ | ✓ | ✓ | ✓ | ✓ | ✓ |

However, as shown in Tab. 1, current EQA research predominantly considers the household scenarios (like kitchens or rooms) by utilizing simulated household environments (*e.g.*, House3D [45], AI2-THOR [15], Habitat [25, 38, 32]). These tasks usually involve general understanding tasks like identifying objects, attributes, or spatial relationships, *etc.*, requiring short phrasal responses. While these benchmarks are instrumental in advancing foundational capabilities, they still exhibit certain limitations. Firstly, their focus on specific indoor scenarios limits applicability to distinct environments, such as industrial scenarios. Secondly, they primarily emphasize general capabilities, while overlooking specific perspectives like safety within the environment. Thirdly, most of them lack the procedural or causal process after giving the short phrasal answer, which leads to insufficient evaluation of the agent's reasoning ability. These constraints limit the effectiveness of EQA in evaluating agent readiness for real-world applications. From practical angle, directly collecting industrial data from real warehouses raises concerns related to costs, privacy, safety, and liability. It also offers limited controllability and diversity over the various assets within the environment.

To address these shortcomings and steer EQA research towards industrial scenarios, we introduce IndustryEQA (see Fig. 1), the *first* benchmark dedicated to evaluating embodied agent capabilities in the safety-critical context of industrial warehouses. IndustryEQA moves beyond the limitations of existing EQA benchmarks by providing a challenging evaluation focused on industrially relevant perception, multifaceted reasoning, and crucial safety awareness. Our benchmark is built on the NVIDIA Isaac Sim platform [24], which provides high-fidelity physics simulation and supports flexible, diverse warehouse designs. This powerful simulation engine enables the creation of realistic industrial scenarios that include human agents—an element largely overlooked in previous EQA benchmarks. Furthermore, as shown in Fig. 1, we curate question–answer annotations across six categories to assess two crucial abilities: safety awareness and general perception. In addition to evaluating the models' direct answers, we also include reasoning annotations to examine their reasoning capabilities. The overall contributions are summarized as follows:

- We take a significant step forward by extending the household EQA task to an industry-oriented setting, enabling the embodied AI community to evaluate agents' safety-aware understanding and reasoning capabilities;
- We propose the first industrial EQA benchmark **IndustryEQA**, which consists of episodic memory videos collected from the Isaac Sim simulator, along with question–answer annotations spanning six categories, accompanied by reasoning-based answers;
- We design a comprehensive evaluation framework and assess multiple baseline models on IndustryEQA. A detailed analysis of the results is provided, offering key insights into model performance.

## 2 Related Work

**Embodied Question Answering (EQA).** The EQA task was first introduced by Das *et al.* [7], which is defined as requiring an agent to actively explore the environment and perform atomic actions based on egocentric visual perception. Later studies, say ScanQA [2], RoboVQA [36], and OpenEQA [23], extend this active setting to episodic memory one, which collects all the visual information and then perform question answering based on the visual perception. Such a setting is similar to visual question answering (VQA), where a model is required to answer natural language questions given visual inputs such as images or videos. A key distinction is that EQA tasks primarily focus on the egocentric view and are not limited to a single question type. Our IndustryEQA takes the episodic memory setting, which is easy to benchmark current visual large language models (VLLMs) [19]. Due to concerns regarding cost, privacy, controllability, and diversity, most existing EQA benchmarks rely on simulators to generate environments. However, almost all of these environments focus on home scenarios like kitchens or rooms, leaving domains like industrial settings largely underexplored. Our proposed IndustryEQA aims to fill this gap within the EQA community.

**Industrial Question Answering.** Robotics has received great attention in recent years, with a broad range of applications actively being explored. One particularly promising domain lies in industrial settings, such as warehouse robotics, which offer substantial practical value. Some initial efforts have begun to bridge industrial settings with natural language understanding. For example, researchers have introduced domain-specific datasets covering various industrial scenarios, including coal mining [34] and customer-driven IT troubleshooting [47], both of which require specialized knowledge to answer. Furthermore, some studies have extended industrial question answering to other modalities, such as audio (*e.g.*, FaultGPT [4]) and video (*e.g.*, QA-TOOLBOX [26]). Among these, QA-TOOLBOX [26] is closely related to our work, as it introduces a data augmentation pipeline built on the manufacturing dataset Assembly101 [35]. In contrast, our proposed IndustryEQA focuses on warehouse environments generated using simulators like Isaac Sim, with a particular emphasis on perception and safety-related understanding. Despite these advances, research on industrial question answering remains limited and underexplored.

## 3 IndustryEQA Benchmark

In this section, we introduce the collection of industry data, including both videos and annotations.

### 3.1 IndustryEQA Simulation

**Simulation Environment.** We use the Isaac Sim simulator developed by NVIDIA[3] to design warehouse scenes and collect simulation data. Our choice of Isaac Sim is motivated by two key advantages. First, it offers high-fidelity and diverse assets with support for adaptive editing, as well as a range of tools for collecting video data. Second, it supports various types of robots and humanoids, all of which can be controlled through the

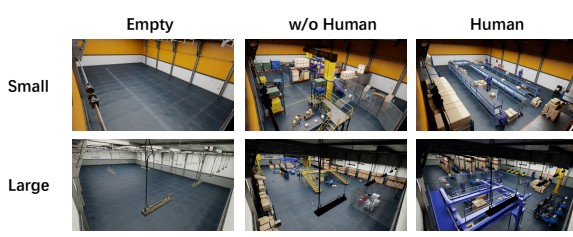

Figure 2: An illustration of industrial scenarios in two sizes (small and large), each comprising three types: empty, without humans, and with humans.

---

[3]https://developer.nvidia.com/isaac/sim

Robot Operating System (ROS). These capabilities make it convenient and suitable to generate various realistic warehouse simulation data.

**Industrial Scenarios and Assets.** To collect industrial video data within warehouse environments, we design a variety of warehouse scenarios in two different sizes: small and large. Specifically, we first create small and large empty warehouses and then populate them with diverse assets to create different scenarios. Isaac Sim simulator supports an expanding library of specialized industrial 3D assets, such as worker models (with animated actions), industrial vehicles, robotic manipulators, conveyor systems, forklifts, pallet racking structures, safety signage, and various inventory types. These complex assets can be used to generate diverse industrial scenarios. We defined safety based on OSHA guidelines, encompassing various types of hazardous energy such as electrical, mechanical, hydraulic, pneumatic, gravitational energy, as well as lock-out/tag-out related issues. In our experiments, we incorporated several default human actions available within Isaac Sim, including walking, organizing, bookkeeping, inspecting, and cleaning. These actions can be further customized according to specific scenarios. Guided by OSHA standards, we positioned human workers intentionally in hazardous scenarios, while placing them randomly in non-hazardous scenarios.

A team of four experts generates 60 small warehouse layouts and 16 large ones. The larger warehouses contain more assets, and the resulting videos are longer and more complex, making them more challenging than the smaller ones. Different from existing EQA benchmarks, our scenarios also incorporate human workers (see Fig. 2), enabling the inclusion of question types related to safety concerns. To design hazardous situations, we refer to the "Hazards and Solutions" documentation[4] from the U.S. Department of Labor, which provides guidance for designing realistic risk scenarios. Further details about this document are provided in the Appendix. Inspired by it, we create dangerous scenes by intentionally placing objects in unsafe configurations, such as falling buckets, toppling boxes, the absence of fire extinguishers, tipped over chemical barrels, placement of electrical equipment near opened liquid containers, unprotected lane sharing between human and motored forklifts, unsecured ladders, visual or physical obstruction of paths and fire-fighting equipments, restrictive entrance and exits preventing effective evacuation, lack of protection against overhead cranes, poor weight balancing on cargo racks, *etc*.

### 3.2 Data Generation Pipeline

The creation of the IndustryEQA benchmark involves a meticulous data generation pipeline (see Fig. 3), encompassing initial video capture from simulated environments, automated Question-Answer (QA) pair generation using advanced Visual Large Language Models (VLLMs), and rigorous human expert refinement. This process ensures the benchmark is high-quality, relevant, and challenging.

**Video Capture.** After establishing the diverse warehouse environments and populating them with industrial assets within Isaac Sim, we utilize Isaac Sim to collect video data. A virtual camera is mounted on the front of a ROS-based Carter robot, simulating an egocentric viewpoint. To enhance the perception field, we set the camera's *z*-axis to 1. Additionally, we configure the camera to record at 30 frames per second (fps) with a resolution of 1080P, ensuring smooth and high-definition video capture. After configuring the camera settings, we collect video data by manually controlling the Carter using the keyboard, aiming to cover every corner of the warehouse during navigation. The captured video is then saved in MP4 format for analysis. It is worth noting that we collect only one video per scenario to ensure diversity across the dataset.

**QA Generation and Refinement.** For each captured episodic memory video, we employ a sophisticated QA generation process leveraging advanced VLLMs. We input each video into the Gemini 2.5 Pro [12] model, guided by a carefully engineered prompt (see Appendix). This prompt instructs the model to generate QA pairs covering the six predefined categories: equipment safety, human safety, object recognition, attribute recognition, temporal understanding, and spatial understanding. A key instruction within the prompt is to ensure a strong emphasis on safety-related questions, making up at least 50% of the generated QAs, and to include questions requiring both simple factual recall and complex reasoning based solely on the video content.

During generation, the VLLM is tasked with producing distinct QA pairs, each consisting of a question, a concise direct answer (a unique, factual response), and a reasoning answer (a single

---

[4]https://www.osha.gov/warehousing/hazards-solutions

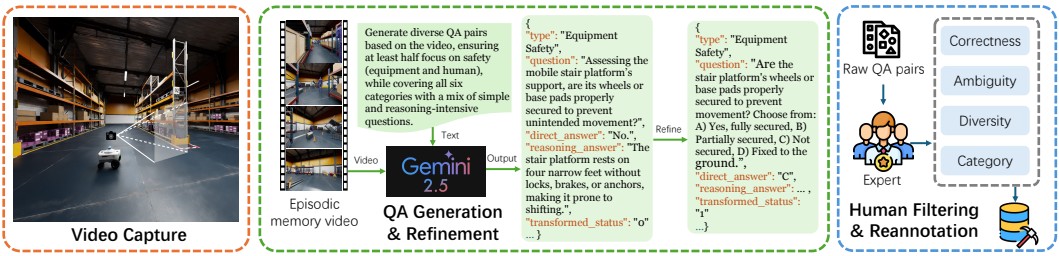

Figure 3: An illustration of data generation pipeline. It consists of three main steps, capturing the video, generating and refining the question answer pairs using an advanced LLM, and finally, having human experts manually filtering out irrelevant pairs and reannotating selected ones.

sentence justifying the direct answer based on visual evidence). The prompt specifies that questions should be unambiguous, vary in difficulty, and avoid referencing specific timestamps or frame numbers. Furthermore, it explicitly requests the generation of complex reasoning process after giving the simple direct answer to thoroughly evaluate the agent's inferential capabilities. This initial phase resulted in a preliminary set of over 3,000 QA pairs. The output is structured in a JSON format, where the reasoning answer is particularly crucial as it provides insight into the expected inferential step, which is vital for evaluating an agent's understanding in our safety-critical industrial context.

To further enhance the benchmark's difficulty and evaluate more nuanced language understanding, a portion of the initially generated QA pairs, especially those with simple "Yes/No" answers, are refined in a subsequent stage. To achieve this, we utilize a combination of VLLMs, including Gemini 2.5 Pro [12], o4-mini, o4-mini-high and o3 [30]. Guided by another specifically designed prompt (see Appendix), the model transformed eligible "Yes/No" questions into open-ended (e.g., "What," "Where," "How") or multiple-choice questions. This transformation is based on the information present in the original direct answer and reasoning answer, ensuring that the refined question format is more challenging while remaining firmly grounded in the video evidence. For example, a question like "Is the worker wearing a helmet?" with a direct answer "No" and reasoning answer "The worker is wearing a baseball cap" might be transformed into "What type of headgear is the worker wearing?". This step aims to increase the complexity of the responses required from the evaluated agents, moving beyond simple binary classifications.

**Human Filtering and Reannotation.** After VLLM-based generation and refinement, all QA pairs are carefully filtered and reannotated by human experts. A team of experts trained in industrial safety and visual data meticulously reviewed each question and its corresponding answers. This critical phase aims to:

1. **Ensure correctness and relevance:** Verify that questions are directly answerable from the video content, answers were accurate, and reasoning is sound.

2. **Filter out ambiguities and errors:** Remove or rectify any QAs that are poorly phrased, ambiguous, or contained factual inaccuracies not caught by the VLLMs.

3. **Enhance challenge and diversity:** Fine-tune questions to increase their inferential depth or nuance, ensuring a good distribution of difficulty levels across all categories. This often involves rephrasing questions to probe deeper understanding or to eliminate unintended shortcuts in reasoning.

4. **Validate categorization:** Confirm that each QA pair is correctly assigned to one of the six predefined categories.

Rigorous human oversight ensures a high-quality, challenging benchmark for evaluating embodied agents' perceptual and reasoning capabilities in industrial EQA tasks. The final dataset comprises 971 QA pairs from small warehouse scenarios and 373 from large warehouse scenarios.

### 3.3 IndustryEQA

**Task Definition.** For the IndustryEQA benchmark, we feed episodic memory videos and questions into emobodied agents to generate answers, then employ an LLM to score both the models' responses

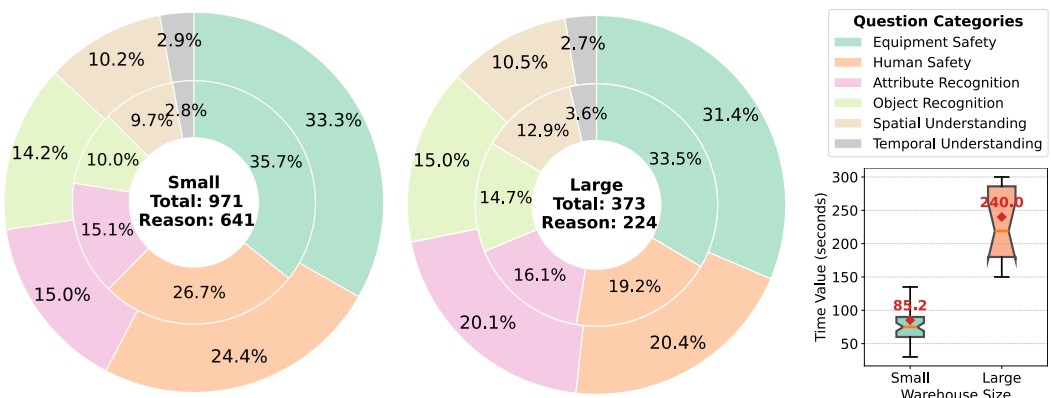

Figure 4: IndustryEQA statistics for small and large warehouses: question category distribution (pie chart) and the time distribution (box plot). The inner ring and outer ring indicate the reasoning and direct QA distribution, respectively. The red dimonds in the box plot denotes the mean time.

and the ground-truth answers. The questions cover two categories: safety and general perception. Safety questions include equipment safety identifying risks associated with warehouse machinery, and human safety evaluating direct hazards to people, such as potential collisions or fall risks. General perception questions span four types: object recognition (identifying objects in the warehouse), attribute recognition (noting characteristics like color or shape), spatial understanding (interpreting positions, distances, and directions), and temporal understanding (sequencing events). A subset of these challenging QA pairs is manually selected and supplemented with reasoning answers.

**Data Statistics.** We provide the statistics of IndustryEQA annotations in Fig. 4. The dataset consists of 76 distinct episodic memory videos (60 from small warehouse layouts and 16 from large ones) paired with the 1344 QA pairs. The figure shows that safety-related QAs dominate (∼50-60%), with equipment safety (∼30%) exceeding human safety (∼20%). Attribute and object recognition together account for about 25–35%, while spatial and temporal understanding make up roughly 10–15%. The inner (reasoning) and outer (direct) rings have nearly identical distributions, confirming that each category's reasoning QAs are collected in proportion to its direct QAs. Moreover, reasoning QAs make up about two-thirds the number of direct QAs. The mean episode duration rises from 85.2 seconds in small warehouses to 240 seconds in large ones, showing that understanding larger environments takes nearly three times longer. The duration in small warehouses ranges from 35 to 120 seconds, whereas in large warehouses it spans 150 to 300 seconds.

**LLM Evaluation Metrics** Following OpenEQA [23], for evaluating the open-vocabulary answers generated by agents, IndustryEQA employs an LLM-based evaluation methodology. For each question $Q_i$, given a human-annotated ground truth answer $A_i^*$ and an agent's generated answer $A_i$, a pre-trained LLM assigns a correctness score $\sigma_i$ on a scale of 1 (incorrect) to 5 (perfectly correct). The aggregate LLM-Match score is then normalized to produce a score (in %) for each answer:

$$C = \frac{1}{N} \sum_{i=1}^{N} \frac{\sigma_i - 1}{4} \times 100\% \tag{1}$$

which providing a quantitative measure of the agent's performance.

## 4 Experiments and Benchmark Results

### 4.1 Baseline Models

To comprehensively assess performance on the IndustryEQA benchmark, we evaluate a diverse set of baseline models based on the OpenRouter API, all employed in a zero-shot setting. These models represent distinct categories based on their architecture and the modalities they utilize.

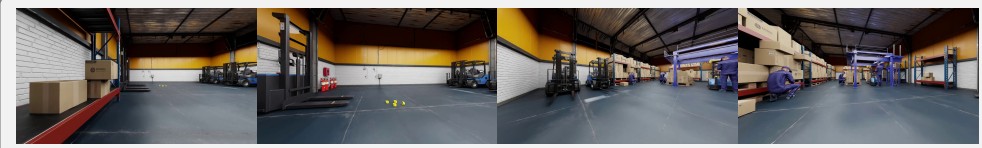

"**question_id**": 364, "**type**": "Human Safety",

"**question**": "What is the spatial situation around the worker operating the pallet jack?",

"**direct_answer**": "Sufficient space."

"**reasoning_answer**": "The worker operating the pallet jack has sufficient space around them, which helps reduce collision risks."

Figure 5: An illustration of the question ID 364 in small warehouse.

**Blind Large Language Models.** This category includes text-only LLMs, which receive only textual questions and must generate answers without any visual context. Their performance indicates how well questions can be answered based on solely prior knowledge or question phrasing alone.

**Multi-Frame VLLMs.** These models are designed to process both visual and textual information. In our benchmark, each model is provided with a textual question along with sampled frames from the corresponding scenario video, enabling them to ground their responses in visual context from the industrial environment. This visual input consists of a sequence of sampled frames, with 30 frames used by default for small warehouses and 40 frames for large warehouses.

**Video VLLMs.** This group consists of VLLMs specifically designed to understand temporal dynamics and extended events within video sequences. They process the textual question alongside the entire video clip (or significant segments), enabling a deeper comprehension of scene context and its evolution over time.

### 4.2  Evaluation Metrics

As shown in Fig. 5, direct answer and reasoning answer are provided in our benchmark. More examples can be seen in the Appendix. Using the scoring mechanism introduced in Eq. (1), we define two primary metrics according to the type of ground-truth answers provided in our benchmark.

**Direct Score (%).** This metric measures the accuracy of the model's direct response. It is calculated using the formula in Eq. (1), where $A_i^*$ is the ground-truth "direct_answer" from our benchmark. This score primarily reflects the model's ability to identify and report factual information accurately.

**Reasoning Score (%).** This metric evaluates the model's ability to provide not only a correct answer but also a clear explanation or reasoning that demonstrates deeper understanding and contextual completeness. It also adopts Eq. (1), but $A_i^*$ corresponds to the ground-truth "reasoning_answer" in our benchmark. This score assesses deeper understanding and reasoning skills, including the ability to grasp spatial relationships, safety implications, or causal connections relevant to industrial environments.

These two primary scores are then systematically analyzed across two critical dimensions, leveraging the detailed annotations and scenario design of our IndustryEQA benchmark.

**Performance by Human Presence.** We compare scores between scenarios with human agents ("Human") and those without ("No Human") to assess how human presence affects model performance, particularly in safety contexts.

**Performance by Warehouse Size.** We also analyze performance differences between "Small" and "Large" warehouses, which vary in complexity and the number of distractors or relevant entities.

### 4.3  Quantitative Results and Analysis

The quantitative findings from our experiments are presented to offer a multifaceted view of model capabilities on the IndustryEQA benchmark.

Table 2: Direct and reasoning answer score performance (%) on the IndustryEQA benchmark: evaluation across human presence scenarios and warehouse sizes.

| Method | Direct Score | | | | Reasoning Score | | | |
| --- | --- | --- | --- | --- | --- | --- | --- | --- |
| | Human Presence | | Warehouse Size | | Human Presence | | Warehouse Size | |
| | Human | No Human | Small | Large | Human | No Human | Small | Large |
| *Blind LLMs* | | | | | | | | |
| GPT-4o-2024-11-20 [28] | 38.10 | 41.68 | 40.06 | 36.53 | 28.67 | 32.18 | 30.54 | 29.52 |
| Gemini-2.0-Flash [10] | 35.99 | 40.88 | 38.67 | 33.38 | 28.67 | 33.06 | 31.01 | 28.21 |
| DeepSeek-R1 [8] | 37.81 | 40.51 | 39.29 | 33.91 | 27.08 | 30.87 | 29.10 | 27.91 |
| DeepSeek-V3-0324 [9] | 36.10 | 43.98 | 40.42 | 33.18 | 27.83 | 32.84 | 30.50 | 27.51 |
| *Multi-Frame VLLMs* | | | | | | | | |
| LLaMA-4-Scout [27] | 51.25 | 50.99 | 51.11 | 52.80 | 46.25 | 40.18 | 43.02 | 42.01 |
| Qwen2.5-VL-72B [3] | 52.62 | 55.31 | 54.09 | 53.42 | 44.00 | 47.29 | 45.75 | 40.06 |
| InternVL2.5-78B [5] | 60.71 | 59.73 | 60.17 | 58.58 | 55.00 | 50.44 | 52.57 | 49.60 |
| Claude-3.5-Haiku [1] | 54.10 | 55.31 | 54.76 | 53.22 | 47.08 | 50.15 | 48.71 | 44.18 |
| GPT-4o-2024-11-20 [28] | 57.23 | 57.52 | 57.39 | 61.39 | 51.50 | 49.49 | 50.43 | 46.39 |
| GPT-4.1-2025-04-14 [29] | 63.95 | 63.53 | 63.72 | 66.42 | 60.33 | 52.49 | 56.16 | 55.22 |
| o4-mini-2025-04-16 [30] | **70.22** | **69.22** | **69.67** | 69.03 | **67.58** | **67.82** | **67.71** | **63.25** |
| *Video VLLMs* | | | | | | | | |
| Gemini-2.0-Flash [10] | 56.95 | 59.87 | 58.55 | 65.82 | 38.00 | 38.64 | 38.34 | 54.72 |
| Gemini-2.5-Flash [11] | 65.21 | 68.05 | 66.76 | **70.24** | 60.67 | 59.68 | 60.14 | 61.45 |
| Gemini-2.5-Pro [12] | 72.34 | 79.67 | 77.31 | 75.71 | 67.37 | 76.74 | 71.33 | 66.30 |
| *Human (test on 100 samples)* | | | | | | | | |
| Human | 67.09 | 64.65 | 72.67 | 71.23 | 46.43 | 54.55 | 51.39 | 48.24 |

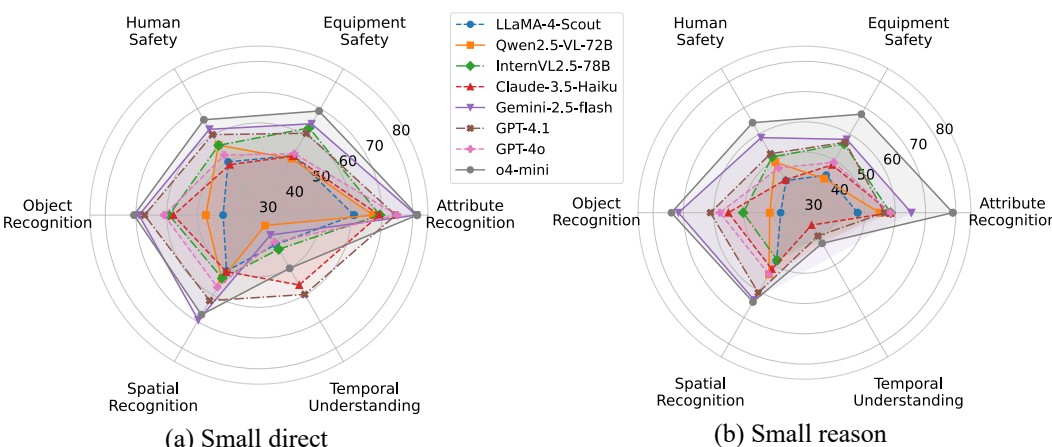

Figure 6: Category-wise performance comparison on the IndustryEQA small-warehouse scenario. (a) and (b) show the direct answer and reasoning answer performance, respectively.

**Comprehensive Performance Overview and Key Insights.** The main results are summarized in Tab. 2, which details the performance of all evaluated baseline models across the key dimensions: Human Presence (Human, No Human), and Warehouse Size (Small, Large). Our experiments on the IndustryEQA benchmark, as detailed in these tables, reveal a clear hierarchy in model capabilities for safety-critical industrial environments. We have the following findings: ❶ **Visual grounding is critical**: Both multi-frame or video-based VLLMs substantially outperform Blind Large Language Models. For instance, leading VLLMs achieve Direct Scores often exceeding 65%, a stark contrast to the performance of Blind LLMs, firmly establishing that visual information is indispensable for accurate perception and response. ❷ **Deeper reasoning remains a hurdle**: Despite the advantages of visual input, a significant challenge persists. Models face difficulty with complex causal, spatial, and temporal understanding crucial for safety awareness when compared to direct factual recall. ❸ **Leading architectures show distinct advantages**: Our annotation model, Gemini-2.5-Pro [12], achieves superior results across all metrics, confirming its high annotation

quality. o4-mini [30], Gemini-2.5-Flash [11], and GPT-4.1 [29] lead among the evaluated models. Notably, the architectural strengths of models like Gemini-2.5-Flash [11] suggest that video understanding aids in complex scenarios by effectively navigating increased environmental intricacy and potential temporal dependencies. ❹ **Safety comprehension reveals systemic challenges**: Further analysis across different conditions shows that models perform comparably on "Human Safety" versus "Equipment Safety" questions but demonstrate considerable room for overall improvement in both domains, suggesting systemic rather than category-specific difficulties. ❺ **Human presence variably impacts models**: Blind LLMs are particularly challenged by the presence of dynamic human agents, whereas top-tier VLLMs often maintain robust performance. However, the intricacies of this VLM interaction with human presence warrant deeper investigation. ❻ **Human annotator performance comparison**: We conducted a simplified experiment to assess human performance by uniformly sampling 100 examples from our benchmark, which were evenly distributed among four annotators. Each annotator meticulously reviewed 25 questions along with their corresponding videos, subsequently crafting direct and reasoning answers. The results indicate that human annotators consistently achieved top-tier performance across almost all Direct Answer scores compared to other VLLMs. However, we noticed that the reasoning answers provided by annotators often reformulated the question into a statement rather than a nuanced reasoning process. Consequently, the evaluated reasoning performance of human annotators slightly underperforms that of certain VLLMs according to our assessment criteria.

**Detailed Category-wise Analysis.** In addition to the overall results, we break down performance across the six core categories in Fig. 6, which shows the category-wise results performance of several VLLM baselines on small warehouse scenrio. The performance on the large warehouse is presented in Appendix. From the results, we draw three key analyses. ❶ **Model perspective**: o4-mini consistently achieves the highest direct-answer and reasoning scores across nearly every category, and Gemini-2.5-flash [11] follow closely behind. ❷ **Category-wise trends**: Attribute Recognition is the easiest task for all models, and temporal understanding is the hardest one. Human Safety and Equipment Safety occupy the middle band, suggesting that the safety-related tasks are challenging. ❸ **Direct v.s. Reasoning**: Almost every model shows a performance drop when moving from direct answers to reasoning answers, highlighting that providing a justification remains substantially harder than simply guessing the correct label.

## 4.4 Ablation Study

To further understand the behavior of the models and the robustness of our evaluation framework, we propose the following ablation study:

**Impact of Sampled Frame Density.** For Multi-Frame VLLMs, the number of sampled frames ($K$) from the input video is a key hyperparameter. To explore this hyper-parameter, we perform an ablation study by varying $K$ from 5 to 50. Fig. 7 shows results for the small-warehouse scenario (large-warehouse results in the Appendix). Increasing $K$ consistently improves both direct and reasoning scores across all

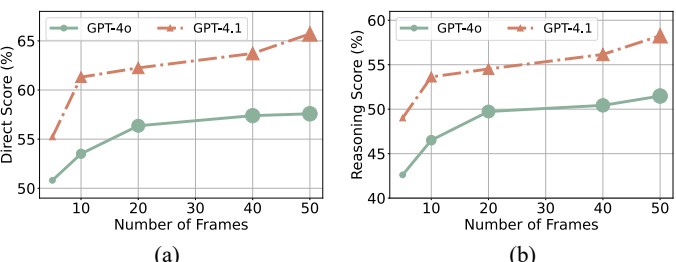

Figure 7: Impact of different sampled frame density w.r.t. (a) Direct Score and (b) Reasoning Score on small warehouse.

VLLMs, but with diminishing returns at higher values, suggesting a potential saturation point. These findings underscore the importance of temporal sampling density for enhancing visual understanding by enriching information coverage.

**Sensitivity to LLM Judge Choice.** Our primary evaluation metrics, Direct Score and Reasoning Score, rely on an LLM judge (*e.g.*, GPT-4o-mini [28]). To assess the robustness of these metrics, we conduct an ablation study using different LLM judges in Fig. 8 by re-evaluating a subset of model outputs using alternative LLMs. This measures how much scores vary by judge and ensures our results hold across different evaluators. From Fig. 8, both judges exhibit similar scoring trends across VLLMs

Table 3: Comparison of direct and reasoning ICC scores across different models of two LLM judges, *i.e.*, GPT-4o-mini and Gemini-2.0-flash.

| Model | Direct Score (ICC) | Reasoning Score (ICC) |
|---|---|---|
| Qwen2.5-VL-72B | 0.953 | 0.925 |
| Claude-3.5-haiku | 0.923 | 0.886 |
| Gemini-2.5-flash | 0.915 | 0.837 |
| o4-mini | 0.903 | 0.873 |

for direct and reasoning metrics. Notably, gpt4o-mini is more lenient on direct scores compared to Gemini-2.0-flash, whereas Gemini-2.0-flash applies a more lenient standard to reasoning scores.

**Reliability of LLM-based Scoring.** To further strengthen our robustness analysis (Fig. 8), we conducted a rigorous inter-rater reliability evaluation between our two LLM judges (GPT-4o-mini and Gemini-2.0-flash). Specifically, we employed the Intraclass Correlation Coefficient (ICC)—a widely accepted statistical metric for quantifying the consistency of quantitative ratings across different raters. Unlike simple correlation coefficients, ICC assesses absolute agreement, making it particularly appropriate for ordinal ratings such as our 1–5 point scoring scheme.

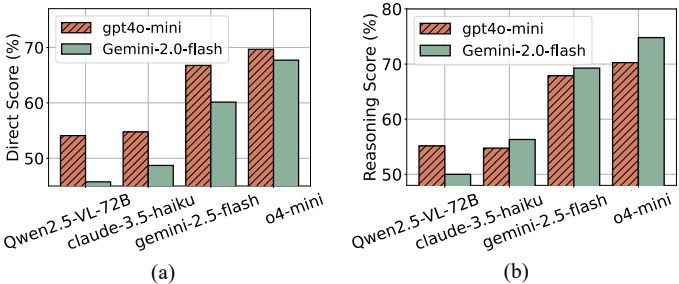

(a)      (b)

Figure 8: Sensitivity to LLM judge w.r.t. (a) Direct Score and (b) Reasoning Score on small warehouse.

As summarized in the table below, our results show consistently high ICC values (ranging from 0.837 to 0.953) across all evaluated models for both Direct and Reasoning scores. These values indicate a strong level of inter-rater reliability, demonstrating that despite minor differences in scoring tendencies, our LLM judges maintain a high degree of agreement in their absolute assessments. This provides solid quantitative evidence for the robustness and reliability of our evaluation framework, confirming that its conclusions are not sensitive to the specific choice of LLM judge. We will include this analysis in our updated version.

## 5 Conclusion

In this paper, we extend the household EQA task to an industry-oriented setting and present the *first* industrial EQA benchmark for the embodied AI community. Leveraging the Isaac Sim platform, we collect episodic-memory videos and generate 1,344 QA pairs spanning six categories—some enriched with reasoning answers. We then introduce a comprehensive evaluation framework to compare multiple baseline models on IndustryEQA and extract key insights from the results. We hope IndustryEQA will offer a fresh perspective for research in embodied AI. In the future, we aim to expand to various scenarios and increase extra modalities like audio or depth map within warehouse. It is also worth trying to expand our episodic memory setting to an active one, and perform supervised training or reinforcement learning.

## Acknowledgement

Tianlong Chen is partially supported by Cisco Faculty Award.

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

## A More Statistics of the Benchmark

### A.1 Word counts of the warehouse annotations

Word counts of the small warehouse annotations and large warehouse annotations are presented in Fig. 9 and Fig. 10, respectively. From the results, we can see that the distribution of words in questions, direct answers and reasoning answers under small and large warehouses is diverse. Notably, question word frequencies are more uniformly distributed than those in direct or reasoning answers, highlighting the greater diversity of the questions.

### A.2 Category-wise performance comparison on the IndustryEQA large-warehouse

The detailed, category-wise performance on the large warehouse is shown in Fig. 12. From these results, we can draw a few observations. First, the object-recognition and attribute-recognition tasks are the easiest in the large-warehouse setting, while the remaining four tasks exhibit comparable levels of complexity. Second, o4-mini and Gemini-2.5-flash achieve nearly identical top-tier performance on the large-warehouse benchmark. Gemini-2.5-flash excels at object recognition and attribute recognition, whereas o4-mini outperforms on equipment safety, spatial reasoning, and temporal understanding tasks. In contrast, Qwen2.5-78B lags behind particularly on temporal understanding. Third, reasoning tasks are noticeably more challenging than direct-answer generation in the large-warehouse setting.

### A.3 Impact of different sampled frame density on large warehouse

The impact of sampled frame density on the large warehouse is illustrated in Fig. 13. As shown in the figure, increasing the number of sampled frames consistently improves the performance of both models across the two metrics. This highlights the importance of temporal sampling strategies in enhancing visual understanding through enriched information coverage.

## B Visualization

Some visualization examples of the IndustryEQA benchmark (including some sampled frames and its corresponding question answer pair) are illustrated in Fig. 11.

## C Question Answer Generation Details

### C.1 QA Generation

Initial question-answer (QA) pairs were generated from videos using Gemini 2.5 Pro, guided by a safety-focused prompt. This prompt directed the model to cover six categories (Human Safety, Equipment Safety, Spatial Understanding, Temporal Understanding, Object Recognition, Attribute Recognition), ensure at least 50% safety-related QAs, and provide distinct direct and reasoning answers in JSON format. This phase produced over 2,000 QA pairs.

### C.2 QA Transformation

A subset of "Yes/No" QA pairs was refined using VLLMs (including Gemini 2.5 Pro and o4-mini). These models transformed eligible questions into open-ended or multiple-choice formats to increase complexity, based on the original direct and reasoning answers.

### C.3 QA Refinement

Generated and transformed QA pairs were then evaluated by an LLM. This refinement step assessed QA pairs against criteria including video dependence, type consistency, answerability from the video,

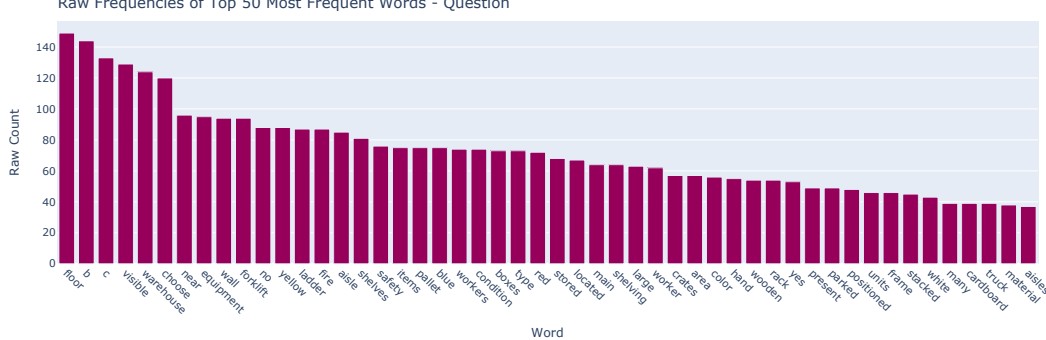

(a) Counts of the top 50 words in the small warehouse questions.

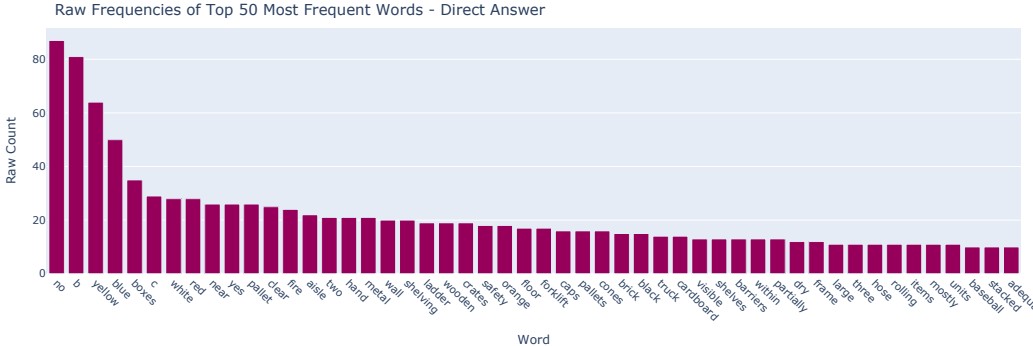

(b) Counts of the top 50 words in the small warehouse direct answers.

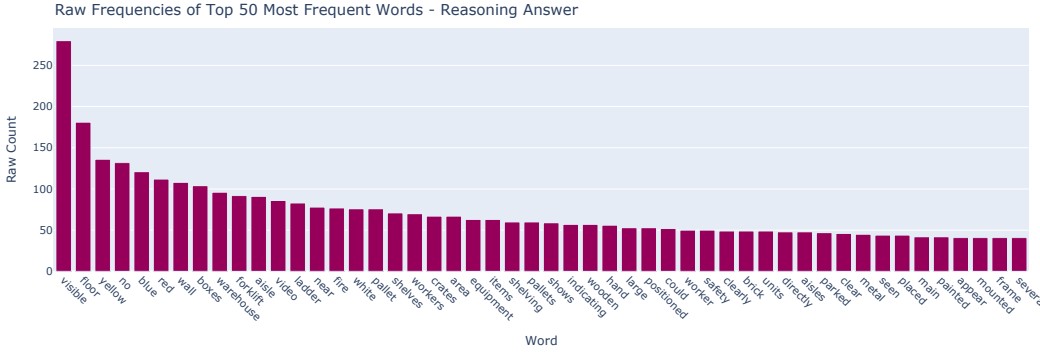

(c) Counts of the top 50 words in the small warehouse reasoning answers.

Figure 9: Distribution of the top 50 word frequencies in the small-warehouse QA data (from top to bottom: questions, direct answers, and reasoning answers).

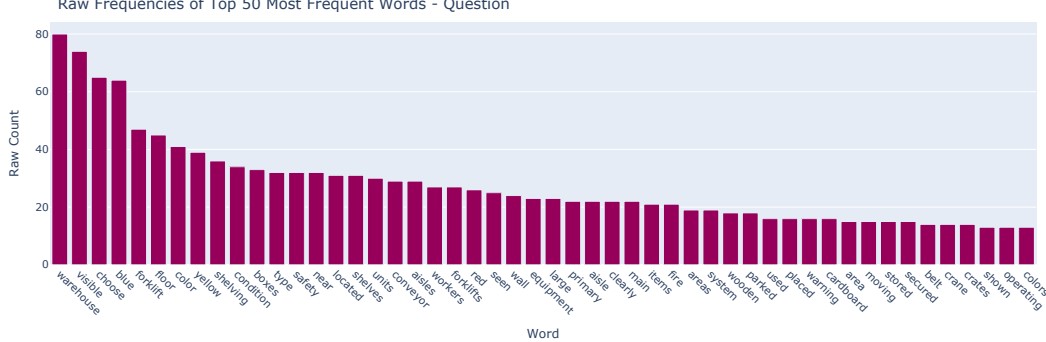

(a) Counts of the top 50 words in the large warehouse questions.

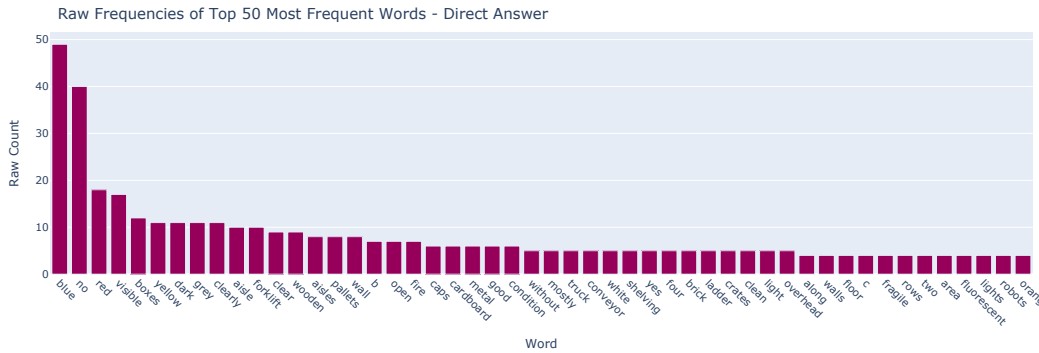

(b) Counts of the top 50 words in the large warehouse direct answers.

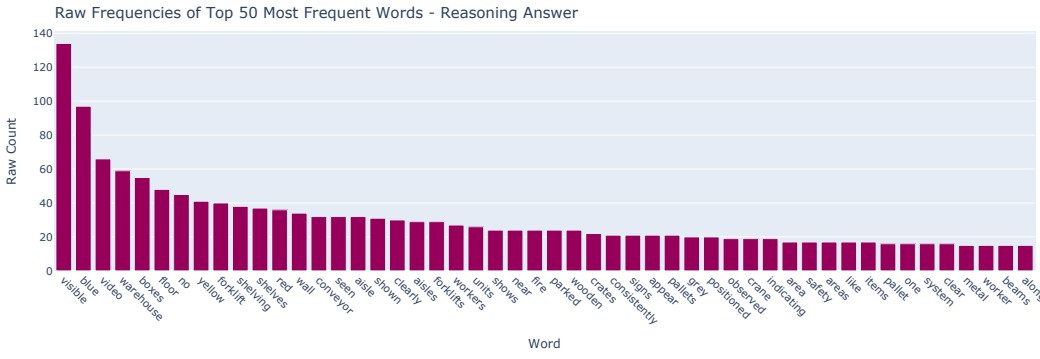

(c) Counts of the top 50 words in the large warehouse reasoning answers.

Figure 10: Distribution of the top 50 word frequencies in the large-warehouse QA data (from top to bottom: questions, direct answers, and reasoning answers).

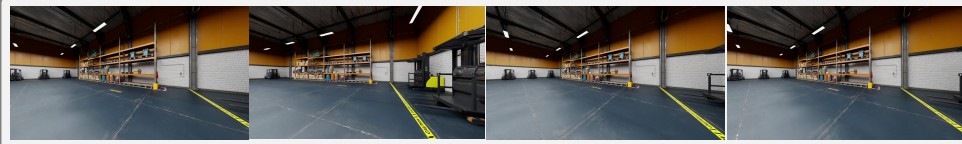

"**question_id**": 658, "**type**": "Equipment Safety",

"**question**": "Concerning the overhead utilities, which statement is correct? (A) All cables and pipes are enclosed in conduit, (B) Some electrical cables are exposed and hanging, (C) No overhead utilities are visible?",

"**direct_answer**": "(B) Some electrical cables are exposed and hanging."

"**reasoning_answer**": "One side of the aisle is bounded by a racking structure and stacked pallets with very little buffer space, while the opposite side remains open."

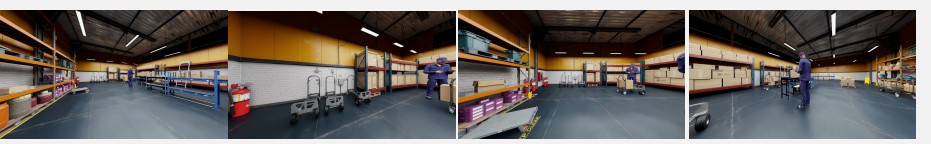

"**question_id**": 686, "**type**": "Object Recognition",

"**question**": "Which of the following pieces of equipment is NOT present in the scene? A) Hand trucks (dollies), B) Ladder, C) Pallet jack, D) Forklift?",

"**direct_answer**": "D) Forklift."

"**reasoning_answer**": "Hand trucks and a ladder are clearly visible, but there is no forklift machinery in any part of the visible area."

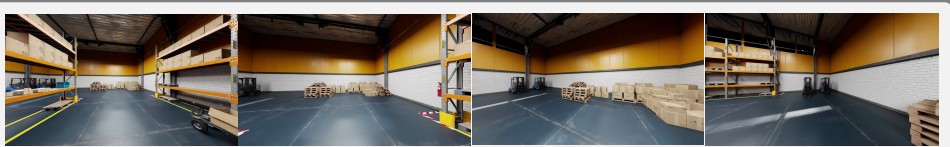

"**question_id**": 709, "**type**": "Attribute Recognition",

"**question**": "What is the dominant colour of the parked industrial vehicle on the far-right side of the frame?",

"**direct_answer**": "Blue (with black lift-mast and forks)."

"**reasoning_answer**": "The body panels of the forklift are clearly painted a bright blue, while only the mast and forks are black, making blue the dominant visible colour."

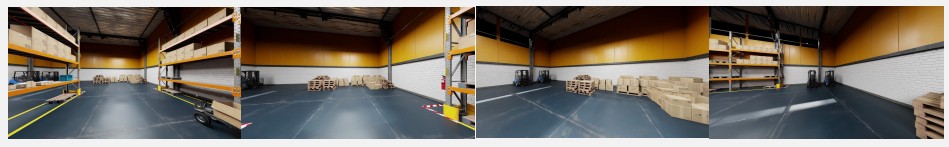

"**question_id**": 705, "**type**": "Equipment Safety",

"**question**": "What is the dominant colour of the parked industrial vehicle on the far-right side of the frame?",

"**direct_answer**": "Blue (with black lift-mast and forks)."

"**reasoning_answer**": "The body panels of the forklift are clearly painted a bright blue, while only the mast and forks are black, making blue the dominant visible colour."

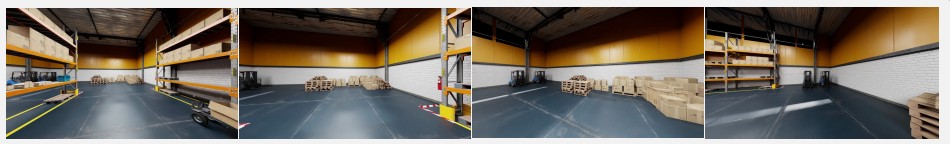

"**question_id**": 711, "**type**": "Object Recognition",

"**question**": "Which of the following objects are positioned in the extreme back-corner of the warehouse? Choose ALL that apply: (A) Stacked wooden pallets, (B) Blue forklifts, (C) Cardboard shipping boxes, (D) Cylindrical metal drums?",

"**direct_answer**": "A, B, C."

"**reasoning_answer**": "The corner area contains neat piles of timber pallets, two idle blue forklifts, and several cardboard cartons on pallets; there are no cylindrical drums visible anywhere in the scene."

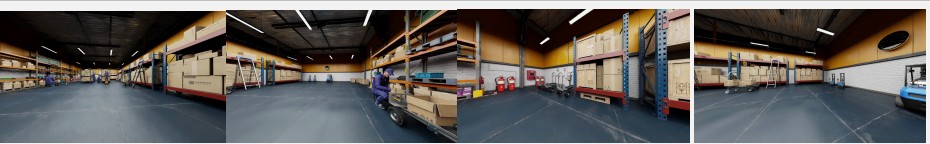

"**question_id**": 736, "**type**": " Attribute Recognition ",

"**question**": " What is the **material** of the clear warehouse floor—polished concrete or wooden planks?",

"**direct_answer**": "Polished concrete."

"**reasoning_answer**": " The floor shows continuous joint lines and a reflective, slightly mottled texture characteristic of sealed concrete, not wood grain."

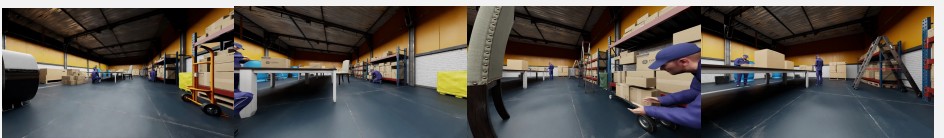

"**question_id**": 757, "**type**": " Causal Reasoning ",

"**question**": " If the mobile ladder in the centre aisle remains unfolded while workers push hand-trucks of cartons toward the loading bay, what is the most likely consequence? (A) A collision that scatters boxes into the walkway, (B) A delay because workers must detour around the ladder, (C) Damage to the ladder's wheels only, (D) No significant effect on workflow?",

"**direct_answer**": "A."

"**reasoning_answer**": " If the mobile ladderBecause the ladder protrudes directly into the main traffic lane, a fully loaded hand-truck has limited manoeuvring space. The momentum of the cart combined with the narrow clearance makes a collision highly probable, and stacked cartons are top-heavy; impact would topple them, obstructing the aisle."

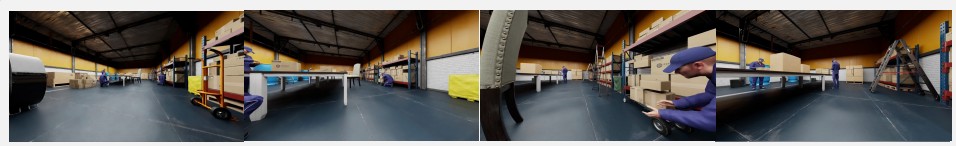

"**question_id**": 759, "**type**": "Object Recognition",

"**question**": "What kind of seating furniture is unusually placed at the worktable?",

"**direct_answer**": "Upholstered dining chair."

"**reasoning_answer**": "A tall, light-coloured cushioned backrest with decorative studs resembles a household dining chair, contrasting with typical industrial stools."

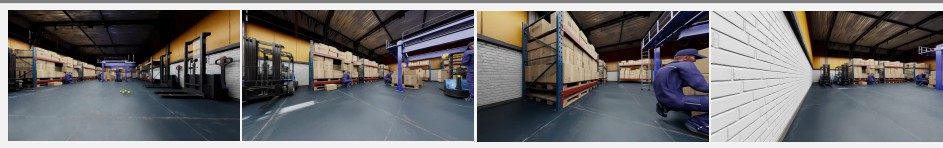

"**question_id**": 762, "**type**": "Human Safety",

"**question**": "Is the crouching worker at center using a neutral spine and bent-knee lifting posture when handling the carton??",

"**direct_answer**": "No."

"**reasoning_answer**": "The worker's back is rounded and the knees are sharply flexed in a deep squat, indicating awkward stooping rather than the recommended power-lift stance."

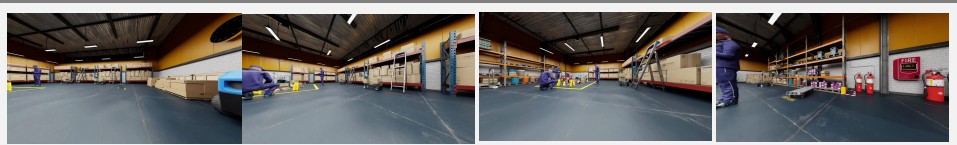

"**question_id**": 788, "**type**": "Equipment Safety",

"**question**": "What primary risk is posed by the unattended flat-bed trolley left length-wise in the main aisle?",

"**direct_answer**": "Collision with moving vehicles or pedestrians."

"**reasoning_answer**": "The trolley narrows the aisle width and lacks any chocks or brakes, so a shallow impact could propel it into a forklift's path."

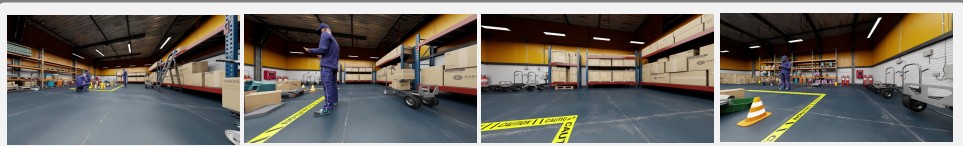

"**question_id**": 793, "**type**": "Causal Reasoning",

"**question**": "Is the cordoned-off area with cones and tape likely a result of routine scheduled maintenance or an unexpected incident creating a potential hazard?",

"**direct_answer**": "An unexpected incident creating a potential hazard."

"**reasoning_answer**": "The presence of what appears to be spilled or damaged goods, along with the reactive response of multiple workers, suggests an unplanned event requiring hazard control."

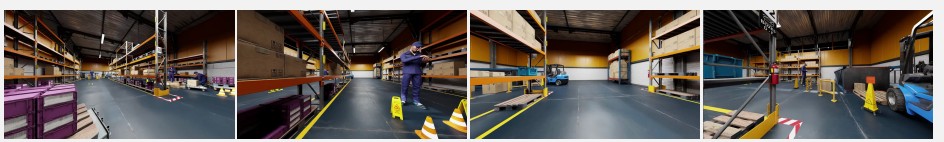

"**question_id**": 841, "**type**": "Spatial Understanding",

"**question**": "Which direction does the central aisle run relative to the camera?",

"**direct_answer**": "From front to back."

"**reasoning_answer**": "The aisle leads straight away from the camera toward the far end of the warehouse."

Figure 11: Examples of IndustryEQA benchmark QA paris.

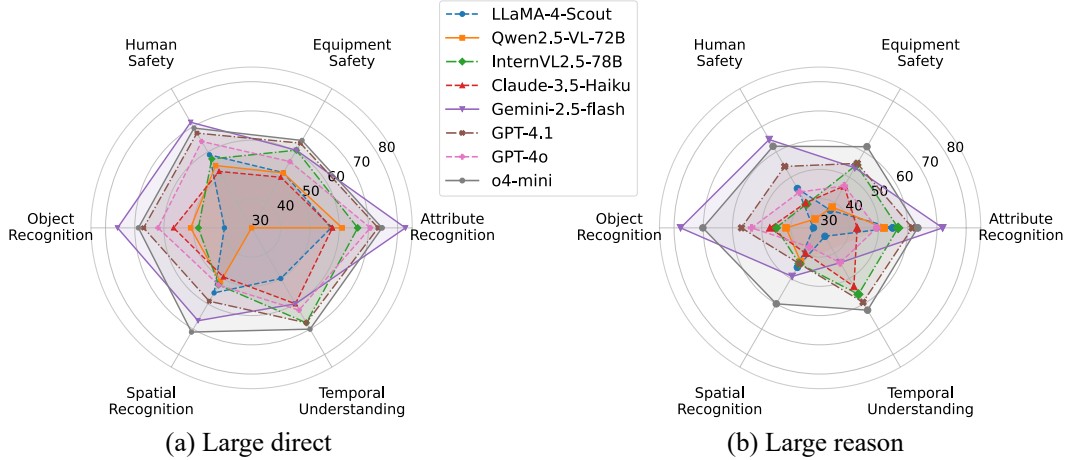

(a) Large direct                    (b) Large reason

Figure 12: Category-wise performance comparison on the IndustryEQA large-warehouse scenario. (a) and (b) show the direct answer and reasoning answer performance, respectively.

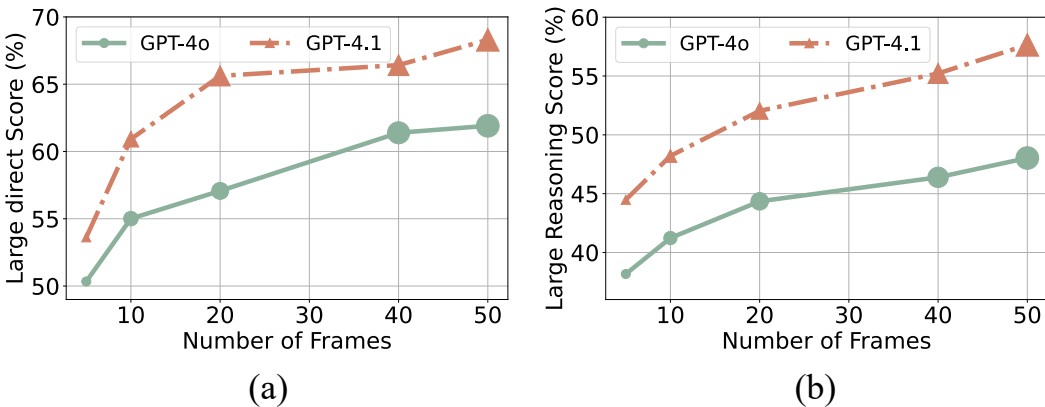

(a)                                 (b)

Figure 13: Impact of different sampled frame density w.r.t. (a) Direct Score and (b) Reasoning Score on large warehouse.

and correctness of both direct and reasoning answers, providing a retain/remove flag and suggesting corrections.

### C.4 Human Filtering

Finally, all QA pairs underwent a meticulous review by human experts. This ensured correctness, relevance, and clarity, filtered errors, enhanced the challenge and diversity of questions, and validated category assignments. The process yielded the final dataset of 971 QA pairs for small warehouses and 373 for large warehouses.

## D Experiments

### D.1 Baseline Model Setup

All baseline models were evaluated in a zero-shot setting via the OpenRouter API. Vision-Language Models (VLMs) were prompted to provide direct and reasoning answers in a specified JSON format. Multi-Frame VLLMs received textual questions and sampled video frames (30 for small warehouses, 40 for large, by default). Video VLLMs processed questions with entire video clips or segments. Blind LLMs received only textual questions and were prompted to answer based on common warehouse knowledge.

## D.2 Evaluation Protocol

Model responses were assessed using Direct Scores and Reasoning Scores, calculated via an LLM judge (GPT-4o-mini and Gemini-2.0-flash). A 1-5 scale was used, with scores normalized to a percentage as shown in the main paper. The LLM judge evaluated direct answers based on one prompt and reasoning answers based on a separate prompt, which included instructions to penalize generated reasoning if the corresponding direct answer fundamentally contradicted the ground truth.

## D.3 Scene Density Impact

We conducted an analysisn (see Tab. 4) to investigate the correlation between model performance and scene density, defined by counting specific objects present in each video (e.g., fire extinguishers, forklifts, shelves, workers, etc.). The table below presents a representative subset of our findings across all videos. Based on these results, we did not identify a clear correlation between the number of specific objects and the direct or reasoning scores. This lack of correlation may be attributed to the complexity and entanglement of multiple object types within the scenes.

| Warehouse Size | Scene | Fire Extinguishers | Forklift | Shelves | Workers | Direct | Reasoning |
|---|---|---|---|---|---|---|---|
| Small | no_human_1 | 0 | 4 | 10 | 0 | 55.26 | 56.58 |
| Small | no_human_2 | 0 | 2 | 15 | 0 | 70.83 | 88.89 |
| Small | no_human_3 | 4 | 1 | 15 | 0 | 81.82 | 81.82 |
| Small | no_human_4 | 2 | 3 | 15 | 0 | 87.50 | 75.00 |
| Small | no_human_5 | 0 | 0 | 4 | 0 | 85.42 | 83.33 |
| Small | no_human_6 | 0 | 1 | 10 | 0 | 51.39 | 65.28 |
| Small | human_1 | 0 | 4 | 10 | 7 | 54.55 | 56.82 |
| Small | human_2 | 1 | 1 | 10 | 7 | 85.00 | 95.00 |
| Small | human_3 | 0 | 2 | 14 | 8 | 53.57 | 51.79 |
| Small | human_4 | 6 | 2 | 14 | 4 | 67.19 | 70.31 |
| Large | no_human_1 | 0 | 2 | 16 | 0 | 62.50 | 69.23 |
| Large | no_human_2 | 4 | 6 | 29 | 0 | 72.66 | 78.91 |
| Large | no_human_3 | 5 | 6 | 29 | 0 | 63.24 | 66.91 |
| Large | no_human_4 | 8 | 2 | 32 | 0 | 82.29 | 91.67 |
| Large | human_1 | 0 | 2 | 11 | 16 | 66.38 | 76.72 |
| Large | human_2 | 0 | 2 | 16 | 14 | 78.12 | 84.38 |
| Large | human_3 | 4 | 6 | 29 | 7 | 60.00 | 70.00 |
| Large | human_4 | 8 | 6 | 21 | 8 | 78.70 | 76.85 |
| Large | human_5 | 8 | 2 | 32 | 7 | 58.33 | 68.33 |
| Large | human_6 | 0 | 4 | 12 | 9 | 60.29 | 64.71 |

Table 4: Scene-wise performance across different warehouse sizes, human presence, and object configurations.

# E Future Work

In this work, we have focused on producing realistic scenes that demonstrate a wide variety of hazards *before* or *after* an incidence occurred, without a soundtrack. We aim to expand beyond this limitation in the future through the following lenses:

**Scene Variety Expansion.** We will further expand the variety of scenes and potential hazard types to include visible gas leaks, fluids (*e.g.*, blood, oil, chemicals), active flames, heavily damaged equipment, animated equipments, traveling personnels at various tasks, unconscious and injured workers, more variety in lightning conditions (*e.g.*, colored, inconsistent, flashing), manufacturing machinery and industrial pipelines, assembly and cargo transport robots, cargo trucks loading & unloading, multilingual safety signs, *etc*.

**Audio Track Simulation.** In addition to the existing video, we also plan to add audio tracks for a more realistic setup, for example: fire alarms, vehicle reversing alerts, announcement broadcasts, motor vehicle operational whines, object collision sounds, multilingual dialogues or whispers from human workers, industrial crane operational whines, *etc*. There have been work demonstrating the value of including audio tracks in visual question answering [17, 48], such as asking questions specifically about the audio.

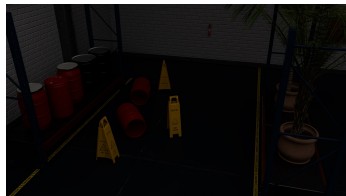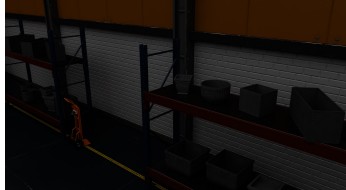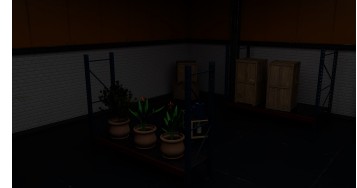

Figure 14: New version under development.

In addition, we believe there is more potential in further scaling up the utility of IndustryEQA and its future versions through designing robust verifiers and reward models, in order to foster reasoning improvements in foundation models.

**Supervised Fine Tuning and Reinforcement Learning.** Due to the controlled simulation nature of the scene building process, we can potentially introduce verifier reward functions on the model outputs, as well as training safety-focused process reward models (PRM, *e.g.*, [20, 40, 21]) and outcome reward models (ORM, *e.g.*, [6, 16, 14]) or setting up as instruction fine tuning [31]. This will help accelerate reinforcement learning research (*e.g.*, Group Relative Policy Optimization [37] and Direct Preference Optimization [33]) in the industrial safety domain. Recent work such as LiFT [42], UnifiedReward [43], LLaVA-Critic [46] and Q-Insight [18] are promising examples of modern multimodal reward model in other settings, some with reasoning justifications [41].

# F  Limitations

**Art Asset Variety.** In this work we started out with a large art asset collection based on Issac Sim, however it can be better. For example, there are only a handful variations of overhead walkways, which can limit the geometrical floor plan layout of the virtual warehouse when articulating vertical utility of space. Labels on various containers are not exhaustively comprehensive, for instance, certain indicator markings of cargo weight class were not available on particular container types. Chemical hazard markings' availability on liquid containers were less than ideal. Storage organization tools such as tie-down straps and ropes were not flexible enough to accommodate a wide variety of container shapes and sizes. These causes certain types of non-OSHA-violation scenarios to be either difficult or impossible to recreate, but ultimately can be resolved through importing more art assets.

**Event Driven Simulation.** The virtual scenes in this work, despite already challenging for many latest open and closed models, can become even more realistic by virtue of event-based simulation. Currently our virtual actors, be it human worker characters, motor-powered forklifts or operating cranes, mostly remain conservative in movements. There is a lack of both spotaneous every-day interpersonal work interaction, a lack of planned and unplanned movements by the virtual workers, and no enduring simulation of extreme events such as earthquakes, acidic rains, wildfires, hurricane or tornado storm, etc. Some of these events happen in the real world and have specialized evacuation protocols[5]. There is also no simulation of time-of-day transitions, nor the swarm patterns of people coming to work or going home, heading to or coming back from lunch, *etc*.

```
Prompt: Safety-Focused QA Pair Generation

Role:  Act as an expert safety analyst for industrial warehouse environments.

Input:  A video recording from a warehouse setting.

Task:  Analyze the video and generate a comprehensive set of relevant
Question-Answer (QA) pairs based only on the video content.  For each QA
pair, generate multiple type-question-answer pairs.  For answer, generate
direct answer and reasoning answer.  Cover all the elements that appear in
the video.

Core Focus & QA Categories:  Your primary goal is generating diverse QA pairs
with a strong emphasis on safety (at least 50% of total QAs).  Cover the
```

[5]https://www.osha.gov/laws-regs/regulations/standardnumber/1910/1910.38

following categories, ensuring a mix of simple factual questions and complex reasoning questions:

1. Safety-Focused Question
   - Human Safety: Evaluating direct risks to human safety, such as potential collisions, falling hazards, ergonomic issues, proper usage of personal protective equipment, and hazardous zones.
   - Equipment Safety: Recognizing risks associated with warehouse equipment, including pathway obstructions, improper stacking, equipment placement, spills, obstacles, inadequate lighting, and fire hazards.
2. General Scene Understanding
   - Spatial Understanding: Questions related to object positions, distances, directions, and spatial relationships.
   - Temporal Understanding: Understanding the sequence and order of events, including counting objects or occurrences over time in videos.
   - Object Recognition: Identifying and classifying objects present in the scenes.
   - Attribute Recognition: Identifying object attributes such as color, size, shape, state, and condition.

QA Requirements:

1. Generate 50 high-quality, distinct QA pairs. Cover all the elements that appear in the video, the richer the better.
2. Prioritize questions with factual, objective answers based on visual observation.
3. Focus on one specific element or condition per question; allow 10% multiple-choice format when suitable.
4. Ensure questions align with their assigned type.
5. Vary difficulty from simple to complex, strictly based on the video.
6. Questions and answers should be concise.
7. For each question, "direct answer" should be unique rather than multiple ambiguous answers, and "reasoning answer" should not exceed one sentence.
8. Do not reference specific times or frames.
9. Ensure clarity and unambiguity in all questions.
10. Although each type of question have a reasoning_answer, you still need to explicitly generate some difficult Causal Reasoning and Commonsense Reasoning type questions.
11. Do not generate questions those whose answers can probably be guessed.

Output Format: Provide your answers in strictly valid JSON format.

Examples:
```
[
{
"id": 1,
"type": "Attribute Recognition",
"question": "What colors are the interlocking, stackable items placed on the floor in aisle 02? Choose the answer from the following options: Yellow, Blue, Green.",
"direct_answer": "Yellow.",
"reasoning_answer": "interlocking, stackable items are placed near workers, which can be seen that they are yellow."
},
{
```

```
"id":  2,
"type":  "Object Recognition",
"question":  "What piece of equipment is used by a worker to move multiple
boxes stacked vertically?",
"direct_answer":  "Hand truck.",
"reasoning_answer":  "A worker is seen using a hand truck to transport
boxes."
},
]

Now analyze the video and generate the QA pairs:
```

## Prompt: Industrial QA Dataset Transformation

Role:  You are an Embodied Expert for refining industrial QA datasets.

Input:  List of JSON QA pairs (keys:  "question", "direct_answer",
"reasoning_answer").

Task:  Analyze each QA pair.  Transform eligible "Yes/No" questions into
open-ended (e.g., "What", "Where", "How") questions or multi-choice questions
based *only* on their direct_answer and reasoning_answer.

Transformation Guidelines:

    1. Eligibility:

          • direct_answer is "Yes." or "No.".
          • Crucial:  The reasoning_answer MUST provide specific descriptive
             information that can form a new question and answer.
              – If "No", reasoning_answer should state *what IS observed*
                instead (e.g., Original Q: "Wearing hard hats?", DA: "No.",
                RA: "Wearing baseball caps." -> Transformable).
              – If "Yes", reasoning_answer should give *specific details*
                supporting the "Yes" (e.g., Original Q: "Aisles clear?",
                DA: "Yes.", RA: "Aisles are unobstructed and wide." ->
                Transformable).  In this case, you should
              – Do NOT transform if reasoning_answer merely confirms the
                "No" (e.g., "No hard hats seen") or is a generic "Yes" (e.g.,
                "Appears correct").
          • You are free to reject to transform it, be objective.  Consider
             whether the original form is more capable of evaluating different
             LLM agents or the modified form.

    2. Transformation Steps (If Eligible):

          • New Question:
              – Based on the specifics in the original reasoning_answer,
                formulate a new question.
              – If the answer is unique and unambiguous, it is much better
                if the question can be changed to open-ended.  Avoid this if
                multiple valid descriptions apply (e.g., an aisle that is both
                'clear' and 'black'.  But if 'clear' is used as the ground
                truth answer, we lose the equally correct detail 'black').
              – If multiple valid descriptions is work for the question,
                tranform it into Multiple Choice:  If the reasoning_answer
                describes a clear state, attribute, or object that can
                be contrasted with a plausible alternative, formulate a
                multiple-choice question.
                 ∗ The question should clearly present concise options.
                  Example format:  "What is the condition of X? Choose from:
                  [Option A], [Option B], [Option C]..." or "Is X [Attribute
                  1] or [Attribute 2]...?".
                 ∗ You should provide at least three options.  One option
                  must directly reflect the information in the original

reasoning_answer. Other options should be relevant
alternatives, and make sure the other options are not right
answers for the question.
- New Direct Answer:
  - This should be the correct option (if multiple-choice) or the
    concise factual answer (if open-ended), directly derived from
    the original reasoning_answer.
- New Reasoning Answer:
  - New reasoning, concisely supporting the new direct_answer.
    It should directly state the factual basis for why the new
    direct_answer is correct, without explicitly meta-referencing
    the transformation process or the "original observation"
    itself. Remember, you can also refer to the video information
    to refine, enrich or correct the reasoning answer.
- Add Key: "transformed_status": "1".

3. Non-Transformed Items:
   - Keep original data.
   - Add Key: "transformed_status": "0".

Examples:

*Example 1: Transformation to Open-Ended*
- Input:
  - "question": "Are the warehouse workers wearing hard hats?"
  - "direct_answer": "No."
  - "reasoning_answer": "The workers visible are wearing baseball
    caps; no hard hats are seen."
- Output: {{
  "question": "What are workers wearing on their heads?",
  "direct_answer": "Baseball caps.",
  "reasoning_answer": "The workers visible are wearing baseball caps,
  no other type of hats are seen.",
  "transformed_status": "1"
  }}

*Example 2: Transformation to Multiple Choice*
- Input:
  - "question": "Are aisles clear?"
  - "direct_answer": "Yes."
  - "reasoning_answer": "Aisles are unobstructed and wide."
- Output: {{
  "question": "What is the condition of the aisles? Choose from:
  Clear, Obstructed.",
  "direct_answer": "Clear.",
  "reasoning_answer": "The aisles are free of obstructions and allow
  passage.",
  "transformed_status": "1"
  }}

*Example 3: No Transformation*
- Input:
  - "question": "Are any tools left on the floor?"
  - "direct_answer": "No."
  - "reasoning_answer": "No tools are visible on the floor."
- Output: {{
  "question": "Are any tools left on the floor?",
  "direct_answer": "No.",
  "reasoning_answer": "No tools are visible on the floor.",
  "transformed_status": "0"
  }}

Output: Strictly valid JSON list of all processed QA objects (original or transformed).

Your turn:
Input:
QUESTION: {question}
ORIGINAL DIRECT ANSWER: {direct_answer}
ORINIGAL REASONING ANSWER: {reasoning_answer}

Output:

## Prompt: EQA LLM-based Refinement

Role: You are an expert evaluator of embodied video question-answering datasets.

Task: Evaluate a question and answer pair (including its assigned type, direct answer, and reasoning answer) based on the warehouse video you've just seen and the generation guidelines.

VIDEO CONTENT: The video shows a warehouse environment.

QUESTION: {question}
ORIGINAL DIRECT ANSWER: {direct_answer}
ORINIGAL REASONING ANSWER: {reasoning_answer}

Evaluation Criteria:

1. QUESTION QUALITY ASSESSMENT:
   - Most important: Video Dependence / Human vs. LLM Distinction: Can the answer be easily guessed using common sense or general warehouse knowledge *without* needing specific details from *this particular* video? High-quality questions require observation of specifics unique to the video. Avoid universal common-sense questions.
   - Type Consistency: Does the question genuinely fit the assigned type?
   - Answerability from Video: Is the question clearly and unambiguously answerable *solely* from the video footage?
   - Relevance: Is the question relevant to the *specific scene* shown (operations, safety, layout, objects)?
   - Specificity, Objectivity & Clarity: Is the question specific, unambiguous, objective, and focused on a single point?

2. ANSWER ASSESSMENT (Direct & Reasoning):
   - Direct Answer Correctness & Conciseness: Is the direct_answer factually correct based *only* on the video? Is it concise and directly responsive?
   - Reasoning Answer Correctness & Format: Does the reasoning_answer accurately explain *how* the direct_answer is derived *from the video*?

Strictness Example (Maintain this):
"question": "Could the open A-frame ladder potentially fall?", "type": "Human Safety", "direct_answer": "Yes.", "reasoning_answer": "Open ladders can be unstable."
*Evaluation Guidance:* Remove (remain: 0). Relies on common sense, not unique video details. Fails Video Dependence.

Evaluation Process:
Before outputting the final JSON response, first provide brief rationales:

1. Retain/Remove Rationale: Briefly explain *why* the QA pair should remain (meets criteria, esp. Video Dependence, Type Match) or be removed (fails criteria).

2. Answer Correctness Rationale: Briefly explain *why* the direct_answer and reasoning_answer are correct or incorrect based *strictly* on video evidence and format requirements.

Then please provide your evaluation in the following JSON format: {{
"remain": 0, // 0 if question should be removed, 1 if it should remain
"direct_answer_correct": 1, // 0 if original direct_answer is incorrect, 1 if correct
"reasoning_answer_correct": 1, // 0 if original reasoning_answer is incorrect/bad format, 1 if correct
"suggested_direct_answer": "Same as original", // Or your corrected direct answer
"suggested_reasoning_answer": "Same as original" // Or your corrected reasoning answer (single, concise, video-based sentence)
}}

**Prompt: Evaluation of Different Agents**

You are an industrial question answering agent tasked with answering
questions about industrial warehouse environments.  You will be given a video
recorded inside a warehouse.  Based *only* on the visual information presented
in the video, answer the following user query concisely.  You must provide a
direct answer and reasoning answer seperatively.

Output JSON Format:
{{
"direct_answer":  "",
"reasoning_answer":  ""
}}

Examples for reference:

Example 1:
User Query:  Are there any visible safety hazards on the warehouse floor?
{{
"direct_answer":  "Yes.",
"reasoning_answer":  "The video shows multiple trip hazards including exposed
cables crossing walkways, packaging materials scattered on the floor."
}}

Example 2:
User Query:  What type of storage system is primarily used in this warehouse?
{{
"direct_answer":  "Pallet racking.",
"reasoning_answer":  "The warehouse uses multi-tier pallet racking systems
throughout most of the visible space, with products stored on standardized
pallets placed on horizontal beams."
}}

User Query:  {question}

Output:

**Prompt: Blind LLM Evaluation for Industrial Warehouse QA**

Output JSON Format:
{{
"direct_answer":
"reasoning_answer":
}}

Examples for reference:

Example 1:
User Query:  Are there any visible safety hazards on the warehouse floor?
{{
"direct_answer":  "Likely yes.",
"reasoning_answer":  "Warehouses commonly have safety hazards such as
forklifts in operation, heavy items that could fall, pallets that might be
sticking out from racks, and occasionally spills or objects on the floor that
could be trip hazards."
}}

Example 2:
User Query:  What type of storage system is primarily used in this warehouse?
{{
"direct_answer":  "Pallet racking.",

```
"reasoning_answer":  "Most industrial warehouses use pallet racking systems
as the primary storage solution because they efficiently maximize vertical
space and allow for organized storage of palletized goods."
}}

Example 3:
User Query:  How many workers are visible in the warehouse?
{{
"direct_answer":  "4 workers.",
"reasoning_answer":  "A typical warehouse operation would have several
workers present at any time, including forklift operators, pickers, and
supervisors.  The most common number would be 2-4 workers visible in a given
section of a warehouse."
}}

User Query:  {question}

Output:
```

## Prompt: Direct Answer Match Evaluation

```
You are an AI assistant who will help me to evaluate the response given the
question and the correct answer.  To mark a response, you should output a
single integer between 1 and 5 (including 1, 5).  5 means that the response
perfectly matches the answer.  1 means that the response is completely
different from the answer.

Example 1:
Question:  Is it overcast?
Ground truth answer:  no
Generated answer:  yes
Your mark:  1

Example 2:
Question:  Who is standing at the table?
Ground truth answer:  woman
Generated answer:  Jessica
Your mark:  3

Example 3:
Question:  Are there drapes to the right of the bed?
Ground truth answer:  yes
Generated answer:  yes
Your mark:  5

Your Turn:
Question:  {question}
Ground truth direct answer:  {ground_direct_answer}
Generated direct answer:  {generated_direct_answer}

Output JSON Format:
{{"direct_score":  }}
```

## Prompt: Reasoning Answer Match Evaluation

```
You are an AI assistant who will help evaluate how well a generated reasoning
answer matches the ground truth reasoning for a given question.

You will evaluate the reasoning answer on a scale of 1-5.  5 means the
generated reasoning accurately reflects the same facts, logic, and overall
conclusion as the ground truth reasoning.  1 means the generated reasoning
presents contradictory facts, logic, or reaches an opposite conclusion
compared to the ground truth reasoning.

Consider both the direct answers and reasoning answers provided when
evaluating the reasoning.  Crucially, if the generated_direct_answer
```

fundamentally contradicts the ground_direct_answer (e.g., 'Yes' vs.
'No', or stating an object is present when it's absent), then the
generated_reasoning is supporting an incorrect conclusion. In such cases,
even if the generated_reasoning discusses similar elements or topics
as the ground_reasoning, it cannot be considered a good match and the
reasoning_score must be low (typically 1, or 2 if there's any marginal,
non-contradictory similarity in how the reasoning is framed despite the
factual error).

Example:

Question: What safety hazards are visible in the warehouse?
Ground truth direct answer: Exposed cables and scattered materials
Generated direct answer: Cables on the floor
Ground truth reasoning: The video shows exposed cables crossing walkways and
packaging materials scattered on the floor creating trip hazards.
Generated reasoning: There are cables running across the floor that could
cause workers to trip.

Output: {{
"reasoning_score": 3
}}

Your Turn:
Question: {question}
Ground truth direct answer: {ground_direct_answer}
Generated direct answer: {generated_direct_answer}
Ground truth reasoning: {ground_reasoning_answer}
Generated reasoning: {generated_reasoning_answer}

Output JSON Format:
{{"reasoning_score": }}

