# OpenReview forum: "IndustryEQA: Pushing the Frontiers of Embodied Question Answering in Industrial Scenarios"
_NeurIPS.cc/2025/Datasets_and_Benchmarks_Track — NeurIPS 2025 Datasets and Benchmarks Track poster_

### Official Review · Reviewer_pTv6 · 2025-06-26

**Rating:** 4
**Confidence:** 3

**Summary:**

This paper introduces a novel Question-Answering (QA) dataset designed for warehouse environments, generated using the existing Isaac Sim Simulator. As the first of its kind for the industrial sector, this dataset is constructed within manually created warehouse settings. The data is collected in a video format and includes corresponding question-answer pair labels. Furthermore, the authors have benchmarked the performance of several commercially available Large Language Models on this dataset, offering initial insights into their capabilities in this specialized domain.

**Additional Feedback:**

1) Are the 3D assets and layouts for the warehouse scenes available for public access? Providing this ground-truth 3D data would be an invaluable resource for future research in spatial understanding and simulation.

2） For future embodied applications where on-device processing is critical, performance efficiency is a key factor. It would be beneficial to include a comparison of the computational cost (e.g., inference time) for each benchmarked method.

3) Could the authors elaborate on the unique QA reasoning challenges that emerge when transitioning from common household environments to more complex warehouse settings?

**Dataset Code Accessibility:**

Yes

**Dataset Code Comments:**

The benchmarking code is available but it is suggested to make the user instructions clearer.

**Ethical Considerations:**

No, there are no or only very minor ethics concerns

**Final Justification:**

After reading authors' feedback, most of my concerns are addressed, I believe this benchmark would beneficial for smarter industrial LLM/MLLM design (though the scope is a little bit limited). Thus, I support borderline accept.

**Limitations Weaknesses:**

1)  What methodology was used to determine the distribution of objects and human workers within the simulated warehouse scenes? It would be insightful to discuss how this distribution might influence dataset bias and the generalization capabilities of models trained on it.

2)  Could you provide more detail on how the motion and positioning of human workers were configured and randomized within the simulations?

3) How is "safety" defined and assessed within the industrial QA framework? In addition, how is this work relates to existing work in video anomaly detection and video surveillance, as this could provide a broader context for your safety-related contributions.

4) Could you clarify the specific methodology used to calculate the reasoning score? Furthermore, for ambiguous questions, particularly those related to safety, what steps were taken to ensure the ground-truth answers comprehensively cover the valid answer space?

6） Was there a specific reason for not including Gemini 2.5 Pro in the benchmark comparisons? Its inclusion could offer a more complete view of the current state-of-the-art.

**Strengths Contributions:**

1) The focus on a warehouse setting makes this dataset particularly impactful, providing a crucial resource for advancing research and development in industrial robotics and automated systems.
2) The open-source availability of the dataset and its accompanying code on Kaggle ensures the work's reproducibility.
3) The paper is well-written and easy to follow.

---

> ### Author Rebuttal · Authors · 2025-07-30
>
> We deeply appreciate Reviewer pTv6 for their constructive feedback and high praises for recognizing our work  for its **novelty**, **the impactful perspective**, **open-source availability** and **insightful experiments**, etc. We address the weaknesses below.
>
> **[W1. Object Distribution]** It’s an insightful point! We don’t explicitly generally design scenes based on distribution of objects. Instead, we take inspiration from many real-world warehouse photos and design localized scenarios based on the OSHA guideline and website [1]. For instance, to articulate lateral stability issues from stacked boxes, we intentionally place containers in unsafe rotation angles; to specify slippery surface hazards, we assemble tipped over, lid-popped, liquid containers of different types paired with slippery danger signs. To further explore the influence of specific object to the performance, we also perform experiments as follows:
>
> | Warehouse Size | scene | fire extinguishers | forklift | shelves | workers | direct | reasoning |
> |:---|:---|:---:|:---:|:---:|:---:|:---:|:---:|
> | Small | no_human_1 | 0 | 4 | 10 | 0 | 55.26 | 56.58 |
> | Small | no_human_2 | 0 | 2 | 15 | 0 | 70.83 | 88.89 |
> | Small | no_human_3 | 4 | 1 | 15 | 0 | 81.82 | 81.82 |
> | Small | no_human_4 | 2 | 3 | 15 | 0 | 87.50 | 75.00 |
> | Small | no_human_5 | 0 | 0 | 4 | 0 | 85.42 | 83.33 |
> | Small | no_human_6 | 0 | 1 | 10 | 0 | 51.39 | 65.28 |
> | Small | human_1 | 0 | 4 | 10 | 7 | 54.55 | 56.82 |
> | Small | human_2 | 1 | 1 | 10 | 7 | 85.00 | 95.00 |
> | Small | human_3 | 0 | 2 | 14 | 8 | 53.57 | 51.79 |
> | Small | human_4 | 6 | 2 | 14 | 4 | 67.19 | 70.31 |
> | Large | no_human_1 | 0 | 2 | 16 | 0 | 62.50 | 69.23 |
> | Large | no_human_2 | 4 | 6 | 29 | 0 | 72.66 | 78.91 |
> | Large | no_human_3 | 5 | 6 | 29 | 0 | 63.24 | 66.91 |
> | Large | no_human_4 | 8 | 2 | 32 | 0 | 82.29 | 91.67 |
> | Large | human_1 | 0 | 2 | 11 | 16 | 66.38 | 76.72 |
> | Large | human_2 | 0 | 2 | 16 | 14 | 78.12 | 84.38 |
> | Large | human_3 | 4 | 6 | 29 | 7 | 60.00 | 70.00 |
> | Large | human_4 | 8 | 6 | 21 | 8 | 78.70 | 76.85 |
> | Large | human_5 | 8 | 2 | 32 | 7 | 58.33 | 68.33 |
> | Large | human_6 | 0 | 4 | 12 | 9 | 60.29 | 64.71 |
>
> Based on these results, we did not identify a clear correlation between the number of specific objects and the direct or reasoning scores. This lack of correlation may be attributed to the complexity and entanglement of multiple object types within the scenes. Nonetheless, we agree this is an insightful direction that merits further exploration in future work.
>
>
> **[W2. Human Workers Details]** Thank you very much for your valuable suggestion! In our experiments, we incorporated several default human actions available within Isaac Sim, including walking, organizing, bookkeeping, inspecting, and cleaning. These actions can be further customized according to specific scenarios. Guided by OSHA standards, we positioned human workers intentionally in hazardous scenarios, while placing them randomly in non-hazardous scenarios. We will include these clarifications in the revised version of our manuscript.
>
> **[W3. Safety Definition]** Thank you for this insightful suggestion! In our work, we defined safety based on OSHA guidelines [1], encompassing various types of hazardous energy such as electrical, mechanical, hydraulic, pneumatic, gravitational energy, as well as lock-out/tag-out related issues. In contrast, safety definitions in anomaly detection studies typically focus on abnormal incidents, like violence or vehicle accidents, detected via surveillance cameras. Furthermore, anomaly detection primarily identifies events that are actively occurring, whereas industrial hazard evaluation requires models to reason about potential dangers that have not yet occurred but could pose risks to human safety. From a VLLM perspective, anomaly detection tasks emphasize detection, while our approach emphasizes reasoning capabilities. We will clearly highlight these distinctions in the updated version of our manuscript.
>
>
> **[W4. Reasoning Score Calculation]** Good point! We rely on a carefully designed prompt for LLM to score answers between prediction and the ground truth, largely surrounding facts, logic, and reasoning. We provide the full details in definition and expanded examples in App. page 15 L118-119. In order to eliminate the ambiguity in questions and answers, we have carefully, exhaustively and manually verified all the Q&A pairs as explained in Sec 3.2 L169-185.
>
> **[W5. Why not Gemini2.5 Pro]** Valuable point! We initially did not include Gemini 2.5 Pro in our benchmark comparison because it was used to generate the initial draft of our Q&A pairs, which we felt could provide an unfair advantage. However, in response to the request for a more comprehensive view of the current state-of-the-art, we have benchmarked it and the results are as follows:
>
> | Method | Direct Score (Human) | Direct Score (No Human) | Direct Score (Small) | Direct Score (Large) | Reasoning Score (Human) | Reasoning Score (No Human) | Reasoning Score (Small) | Reasoning Score (Large) |
> |:---|:---|:---|:---|:---|:---|:---|:---|:---|
> | Gemini-2.5-Pro | 72.34 | 79.67 | 77.31 | 75.71 | 67.37 | 76.74 | 71.33 | 66.30 |
>
> The results indicate that Gemini 2.5 Pro achieves state-of-the-art performance across most evaluation dimensions. We hypothesize this strong performance may be attributed to its potential familiarity with the Q&A generation style. Nevertheless, it serves as a valuable high-water mark for future research on our benchmark.
>
> [1] `https://www.osha.gov/`.
>
> **Additional feedbacks**:
>
> **[A1. Availability of 3D Warehouses]** Good suggestion! We will release the 3D warehouse scenarios in the repository soon.
>
> **[A2. On-device Efficiency]** We think it's a meaningful future work and will add it into the limitation discussion of the manuscript.
>
> **[A3. Unique Challenges]** Great point! During manual inspection of model outputs within the experiment, we empirically observed existing models often underperform due to lack of safety standard understanding, incorrect interpretation of hazardous spatial relationships, recognition of warehouse specific objects, etc. For instance, for equipment safety concerns, the benchmark will include questions like “whether the portable fire extinguishers are correctly mounted according to NFPA guidelines in the warehouse”, which rarely exists in household settings.

---

> > ### Comment · Reviewer_pTv6 · 2025-08-04
> >
> > Thanks for the reply! Most of my concerns are addressed. I am lean on borderline acceptance of this paper.

---

> > > ### Author Response · Authors · 2025-08-04
> > > **Response to Reviewer pTv6**
> > >
> > > Thank you for your kind message and thoughtful feedback! We're truly grateful that our clarifications were helpful and that the revisions addressed your concerns. Your insights played a significant role in helping us improve both the clarity and presentation of our contributions. We sincerely appreciate your support in strengthening the paper.

---

### Official Review · Reviewer_qSpn · 2025-07-01

**Rating:** 5
**Confidence:** 3

**Summary:**

This paper introduces IndustryEQA, the first benchmark for evaluating embodied question answering in safety-critical industrial warehouse scenarios, emphasizing both visual perception and reasoning.
Built on Isaac Sim, the dataset features episodic memory videos and 1,344 QA pairs across six categories, including equipment/human safety and spatio-temporal understanding, with reasoning-based annotations for deeper evaluation.

**Dataset Code Accessibility:**

Yes

**Ethical Considerations:**

No, there are no or only very minor ethics concerns

**Final Justification:**

The additional experiments and analyses have addressed all my concerns and improved the completeness of the work. So I raise my score accordingly.

**Limitations Weaknesses:**

1. The paper does not report how well humans perform on the IndustryEQA benchmark. Including a human baseline would help quantify how far current models are from human-level understanding, and provide a clearer picture of the benchmark's actual difficulty.
2. The paper distinguishes between small and large warehouse environments, but does not analyze how scene density—such as the number of shelves, objects, or human agents—affects model performance. This limits our understanding of how visual clutter or complexity impacts embodied reasoning.
3. The benchmark uses LLM-based scoring to assess model answers, but the reliability of this metric is not thoroughly verified. Although a sensitivity analysis is provided, there is no measurement of inter-rater agreement with human judges or across multiple LLM scorers.

**Strengths Contributions:**

1. This work is the first to extend the EQA task to industrial warehouse scenarios, focusing on safety-critical questions with direct relevance to real-world applications.
2. The dataset is of high quality and carefully designed.
3. The paper is well written and easy to follow.

---

> ### Author Rebuttal · Authors · 2025-07-30
>
> We deeply appreciate Reviewer qSpn for their constructive feedback and high praises for recognizing our work  for its **pioneering effort**, **high quality** and **carefully designed scenarios**, etc. We address the weaknesses below.
>
> **[W1. Human Performance]** Thank you for your insightful suggestion! We conducted a simplified experiment to assess human performance by uniformly sampling 100 examples from our current benchmark, which were evenly distributed among four annotators. Each annotator meticulously reviewed 25 questions along with their corresponding videos, subsequently crafting  direct and reasoning answers. Our findings indicate that human annotators consistently achieved top-tier performance across almost all Direct Answer scores compared to other VLLMs. However, we noticed that the reasoning answers provided by annotators often reformulated the question into a statement rather than a nuanced reasoning process. Consequently, the evaluated reasoning performance of human annotators slightly underperforms that of certain VLLMs according to our assessment criteria.
>
> | Method | Direct Score (Human) | Direct Score (No Human) | Direct Score (Small) | Direct Score (Large) | Reasoning Score (Human) | Reasoning Score (No Human) | Reasoning Score (Small) | Reasoning Score (Large) |
> |:---:|:---|:---|:---|:---|:---|:---|:---|:---|
> | Human | 67.09 | 64.65 | 72.67 | 71.23 | 46.43 | 54.55 | 51.39 | 48.24 |
>
> **[W2. Analysis of Scene Density Impact]** Thank you very much for this valuable suggestion! We conducted an analysis to investigate the correlation between model performance and scene density, defined by counting specific objects present in each video (e.g., fire extinguishers, forklifts, shelves, workers, etc.). The table below presents a representative subset of our findings across all videos. Based on these results, we did not identify a clear correlation between the number of specific objects and the direct or reasoning scores. This lack of correlation may be attributed to the complexity and entanglement of multiple object types within the scenes. Nonetheless, we agree this is an insightful direction that merits further exploration in future work.
>
> | Warehouse Size | scene | fire extinguishers | forklift | shelves | workers | direct | reasoning |
> |:---|:---|:---:|:---:|:---:|:---:|:---:|:---:|
> | Small | no_human_1 | 0 | 4 | 10 | 0 | 55.26 | 56.58 |
> | Small | no_human_2 | 0 | 2 | 15 | 0 | 70.83 | 88.89 |
> | Small | no_human_3 | 4 | 1 | 15 | 0 | 81.82 | 81.82 |
> | Small | no_human_4 | 2 | 3 | 15 | 0 | 87.50 | 75.00 |
> | Small | no_human_5 | 0 | 0 | 4 | 0 | 85.42 | 83.33 |
> | Small | no_human_6 | 0 | 1 | 10 | 0 | 51.39 | 65.28 |
> | Small | human_1 | 0 | 4 | 10 | 7 | 54.55 | 56.82 |
> | Small | human_2 | 1 | 1 | 10 | 7 | 85.00 | 95.00 |
> | Small | human_3 | 0 | 2 | 14 | 8 | 53.57 | 51.79 |
> | Small | human_4 | 6 | 2 | 14 | 4 | 67.19 | 70.31 |
> | Large | no_human_1 | 0 | 2 | 16 | 0 | 62.50 | 69.23 |
> | Large | no_human_2 | 4 | 6 | 29 | 0 | 72.66 | 78.91 |
> | Large | no_human_3 | 5 | 6 | 29 | 0 | 63.24 | 66.91 |
> | Large | no_human_4 | 8 | 2 | 32 | 0 | 82.29 | 91.67 |
> | Large | human_1 | 0 | 2 | 11 | 16 | 66.38 | 76.72 |
> | Large | human_2 | 0 | 2 | 16 | 14 | 78.12 | 84.38 |
> | Large | human_3 | 4 | 6 | 29 | 7 | 60.00 | 70.00 |
> | Large | human_4 | 8 | 6 | 21 | 8 | 78.70 | 76.85 |
> | Large | human_5 | 8 | 2 | 32 | 7 | 58.33 | 68.33 |
> | Large | human_6 | 0 | 4 | 12 | 9 | 60.29 | 64.71 |
>
> **[W3. Reliability of LLM-based Scoring]** We agree that a quantitative verification of our metric's reliability is crucial. To supplement our existing robustness check (Fig. 8), we have performed a rigorous inter-rater reliability analysis between our two LLM judges (GPT-4o-mini and Gemini-2.0-flash).
> For this analysis, we use the Intraclass Correlation Coefficient (ICC), a standard statistical measure for assessing the consistency of quantitative ratings given by different judges. Unlike simple correlation, ICC evaluates the absolute agreement among raters, making it particularly suitable for ordinal data like our 1-5 point scores. An ICC value ranges from 0 (no reliability) to 1 (perfect reliability).
> As shown in the table below, our analysis yields consistently good ICC values (ranging from 0.837 to 0.953) across all evaluated models for both Direct and Reasoning scores. This provides strong quantitative evidence that despite minor differences in leniency, our LLM judges exhibit a very high degree of consistency in their absolute assessments. This confirms that our evaluation framework is robust and its conclusions are not dependent on the specific choice of the LLM judge. We will include this experiment in our updated version.
>
> | Model | Direct Score (ICC) | Reasoning Score (ICC) |
> |:---|:---:|:---:|
> | Qwen2.5-VL-72B | 0.953 | 0.925 |
> | Claude-3.5-haiku | 0.923 | 0.886 |
> | Gemini-2.5-flash | 0.915 | 0.837 |
> | o4-mini | 0.903 | 0.873 |

---

> > ### Comment · Reviewer_qSpn · 2025-08-04
> >
> > Thank you for the detailed response. The additional experiments and analyses have addressed all my concerns and improved the completeness of the work. I will raise my score accordingly.

---

> > > ### Author Response · Authors · 2025-08-04
> > > **Response to Reviewer qSpn**
> > >
> > > Thank you for your continued support and thoughtful feedback. We’re delighted that our clarifications addressed your concerns, and your insights have been invaluable in refining both the writing and framing of our contributions. Thanks to your guidance, we believe the paper is significantly stronger.

---

### Official Review · Reviewer_Loa6 · 2025-07-02

**Rating:** 5
**Confidence:** 3

**Summary:**

IndustryEQA introduces the first embodied question-answering (EQA) benchmark specifically for safety-critical industrial warehouse environments, addressing a significant gap in current EQA research that largely focuses on household settings. Built on NVIDIA Isaac Sim, the benchmark features high-fidelity episodic memory videos depicting diverse industrial assets, dynamic human agents, and hazardous situations inspired by real-world safety guidelines. It comprises 1,344 meticulously annotated question-answer pairs across six categories, including critical safety aspects (equipment and human safety), object/attribute recognition, and temporal/spatial understanding, with an emphasis on reasoning capabilities. The data generation pipeline is a hybrid approach combining large language model (LLM) generation (Gemini 2.5 Pro) with rigorous human expert refinement. The paper also presents a comprehensive evaluation framework with various video-language model (VLLM) baselines. While the benchmark stands out for its novelty and strong focus on industrial safety, concerns regarding the immediate accessibility of code and data, as well as the inherent limitations of simulation realism, require attention for its full acceptance.

**Additional Feedback:**

N/A.

**Dataset Code Accessibility:**

Yes

**Dataset Code Comments:**

The authors have provided the data link, code repo, and a detailed readme.

**Ethical Comments:**

This paper introduces IndustryEQA, the first embodied question-answering (EQA) benchmark specifically for safety-critical industrial warehouse environments. It has no or only very minor ethical concerns.

**Ethical Considerations:**

No, there are no or only very minor ethics concerns

**Final Justification:**

My concerns have been well solved.

**Limitations Weaknesses:**

- While Isaac Sim provides high-fidelity simulation, the synthetic nature of the hazards may inherently lack the full complexity, unpredictability, and nuanced sensory cues (e.g., fluid spills, subtle audio cues, unpredictable human-worker interactions) present in real-world industrial environments.
- LLM Dependency in QA Generation: Although human refinement is applied, the initial heavy reliance on Gemini 2.5 Pro for generating QA pairs could introduce subtle biases or limitations, especially concerning the diversity and cultural/regional assumptions embedded within safety scenarios. While the authors state human refinement, a more detailed analysis of potential LLM-introduced artifacts in the Q&A distribution would strengthen the paper.
- The current evaluation omits comparisons with existing industrial SOTA (State-of-the-Art) methods or relevant industrial AI benchmarks (e.g., QA-TOOLBOX, as suggested in the consolidated review). Including such comparisons would strengthen the claims of superiority and provide a more comprehensive understanding of IndustryEQA's positioning within the broader industrial AI landscape.

**Strengths Contributions:**

- IndustryEQA is a pioneering effort as the first EQA benchmark tailored for industrial scenarios, a largely underexplored but highly critical domain in embodied AI. Its strong emphasis on safety-critical aspects (approximately 50% of QA pairs dedicated to equipment and human safety) directly addresses real-world industrial needs and aligns well with the societal impact goals of NeurIPS. The integration of dynamic human agents and OSHA-inspired hazards (e.g., falling objects, obstructed exits) marks a significant advancement beyond static, household-centric environments.
- The benchmark leverages high-fidelity NVIDIA Isaac Sim environments, featuring diverse industrial assets (e.g., forklifts, chemical barrels) and scenarios (small and large warehouses, with and without humans). The hybrid QA generation methodology, combining initial LLM drafting with subsequent meticulous human expert refinement, ensures both scalability and high-quality, relevant annotations, particularly for complex reasoning questions (as illustrated by examples in Figure 3).
- The paper provides a thorough evaluation framework, testing over 15 zero-shot VLLMs (including GPT-4o, Gemini 2.5, o4-mini) across two key metrics: Direct Score (factual accuracy) and Reasoning Score (inferential depth). The comprehensive baseline analysis reveals insightful findings, such as video VLLMs significantly outperforming blind LLMs by over 25%, and persistent challenges in reasoning tasks (Figure 4), highlighting areas for future research.
- The authors describe an LLM-based scoring protocol (Equation 1) and demonstrate its robustness through checks (Figure 8). Furthermore, ablations on frame sampling density (Figure 5) provide valuable insights into temporal comprehension trade-offs, contributing to the understanding and potential reproducibility of results derived from the benchmark.

---

> ### Author Rebuttal · Authors · 2025-07-30
>
> We deeply appreciate Reviewer Loa6 for their constructive feedback and high praises for recognizing our work  for its **pioneering effort**, **high fidelity**, **comprehensive scenarios**, **insightful findings**, and **societal impact potential**, etc. We address the weaknesses below.
>
> **[W1. Synthetic Hazards]** Good point! We have made efforts to enhance both the diversity and realism of the faithfulness hazards in our benchmark. To improve diversity, we have incorporated the majority of available assets provided by Isaac Sim, including multiple types of forklifts, various caution and construction tapes and signs, a wide range of liquid and non-liquid containers, pallets, racks, conveyor belts, etc. Regarding faithfulness, we have meticulously designed hazards according to OSHA safety guidelines  and based on real-world warehouse hazard photos. Examples include violations of minimum aisle width, improper forklift counterbalance, lack of protective rails on stairs or bridges, unsecured stepladder wheels, and lateral stability issues due to stacked boxes.  While it is impossible to cover every possible hazard type, we plan to open source our 3D asset file so that other researchers can contribute and help make it even more comprehensive.
>
> [1] `https://www.osha.gov/`.
>
> **[W2. Q&A Distribution Bias]** Thank you for your insightful suggestion! We have presented our analysis of the question type distribution in Fig. 4, as well as word distributions across questions, direct answers, and reasoning answers in Appendix Fig. 1 and Fig. 2. We acknowledge that our current focus is on US-based safety standards, specifically OSHA. Given any similar rules like OSHA, our framework can produce customized benchmark data according to users' needs. Nevertheless, the overall Q&A distribution is influenced by OSHA guidelines, the range of assets available in Isaac Sim, the visual coverage provided by the camera, and potential biases in VLLMs. Our findings indicate that the Q&A distribution is primarily determined by the first two factors, while the camera and VLLMs have accurately reflected all the information present in each scene.
>
> **[W3. Missing Comparisons]** Thank you for bringing this to our attention! We have provided a comparison of industrial question answering methods in Section 2, Lines 77–89. As emphasized in the paper, our work is the first to introduce a video-based benchmark specifically designed for industrial warehouse environments. Most current state-of-the-art approaches for industrial scenarios rely exclusively on the natural language modality, such as those used in coal mining and customer-driven IT troubleshooting. The only existing video-based industrial question answering method we are aware of is QA-TOOLBOX [1], which is derived from the manufacturing dataset Assembly101 [2]. However, QA-TOOLBOX primarily serves as a data augmentation pipeline for the manipulation and assembly of small objects, rather than for warehouse storage scenarios.
>
> Additionally, we note that the closed-source VLLMs we benchmarked significantly outperform all open-source counterparts (e.g., showing a 15.78% gap between o4-mini and InternVL2.5-78b).
>
>
> [1] R. Manuvinakurike, E. Watkins, C. Savur, A. Rhodes, S. Biswas, G. G. Mejia, R. Beckwith, S. Sahay, G. Raffa, and L. Nachman. Qa-toolbox: Conversational question-answering for process task guidance in manufacturing. arXiv preprint arXiv:2412.02638, 2024.
>
> [2] F. Sener, D. Chatterjee, D. Shelepov, K. He, D. Singhania, R. Wang, and A. Yao. Assembly101: A large-scale multi-view video dataset for understanding procedural activities. In CVPR, pages 21096–21106,2022.

---

### Official Review · Reviewer_vepR · 2025-07-08

**Rating:** 4
**Confidence:** 4

**Summary:**

This work introduces a new EQA benchmark that is focused on industry/warehouse settings as opposed to prior work which has focused almost exclusively on household settings. This has the potential to be an important and timely contribution, expanding the scope/domain of EQA. Environments are generated in Issac Sim, questions are generated by video LLMs and are refined by human experts. In addition to providing the dataset, the work also provides benchmark results of various industry leading models. Models/Agents are evaluated both on "direct" answers as well as "reasoning" answers, which can potentially prevent lucky guesses or hallucinations. Results show that visual grounding and temporal coverage are important and the benchmark isn't yet saturated, showing promise for continued usage.

**Dataset Code Accessibility:**

Yes

**Dataset Code Comments:**

Dataset is accessible through kaggle. Code seems clearly written and largely follows the structure of OpenEQA. I believe it would be appropriate for the authors to more generously attribute OpenEQA in terms of citations and code considering the level of overlap.

**Ethical Considerations:**

No, there are no or only very minor ethics concerns

**Limitations Weaknesses:**

- Engineered dataset in simulation. Unclear how much gap to reality will be there. Scenes, hazardous conditions, videos, and to an extent even questions are all scripted. Gap between this scenario and real-world industry deployment is untested.
- Only episodic memory based evaluation. Considering the scenes are entirely in simulation, not having an active track where the robot has to explore the space feels like a missed opportunity.

**Strengths Contributions:**

- New setting for EQA compared to prior works, providing strong differentiation.
- Reasoning based eval scores in addition to direct eval metrics can better help understand if correctness is due to luck/hallucination vs real understanding.
- Comprehensive baseline and evaluation suite covering 15+ models and ablations

---

> ### Author Rebuttal · Authors · 2025-07-30
>
> We thank Reviewer vepR for the constructive feedback and for recognizing our work as an **important and timely contribution** with **strong differentiation**. We address the weaknesses below.
>
> **[W1. Sim-to-Real Gap]**  Thank you for pointing this out! We acknowledge that our benchmark still exhibits a gap compared to real-world scenarios, as current EQA-based benchmarks primarily rely on simulators. Our simulator, built on Isaac Sim, is among the first to generate highly “vivid scenes,” as recognized by Reviewers Loa6 and qSpn. We believe our work represents a principled and safe step towards real-world deployment. Importantly, the hazards in our simulations are not arbitrarily designed; rather, they are directly inspired by official U.S. OSHA safety guidelines [1], ensuring that our benchmark reflects recognized industrial hazards. We carefully crafted our scenarios based on specific industrial safety concerns (with reference to relevant OSHA safety codes).  Creating a real-world dataset of diverse industrial accidents is infeasible and uncontrollable due to safety, cost, and privacy concerns. Therefore, our high-fidelity simulation serves as an essential and ethical testbed for developing safety-aware AI systems prior to real-world deployment.
>
> [1] `https://www.osha.gov/`.
>
> **[W2. Lack of an Active EQA Track]** Thank you for your constructive comments. Our initial focus on the episodic memory setting was a deliberate choice to establish a controlled and reproducible benchmark for evaluating the core perception and reasoning capabilities of modern VLLMs in maintaining persistent visibility over designated aspects of each scene. For example, we considered factors such as targeted obstructions, occlusions, safety hazards, perceived spatial orientations, and inter-object distances across various camera angles. This setting effectively incorporates dynamic human agents and temporal events, which are critical for industrial safety applications. We fully agree that developing an active track would be a valuable extension, and as noted in our conclusion (Line 330), we consider this a high-priority direction for future work.
>
> **[Code Comments. Attribution to OpenEQA]** Thank you for this important note. In the revised version and code readme, we will add a dedicated acknowledgment to the OpenEQA project and its codebase.

---

### Author Response · Authors · 2025-08-08
**Follow-Up and Clarifications Ahead of Rebuttal Deadline**

Dear Reviewers,

We sincerely appreciate your time, effort, and insightful feedback on our manuscript.

As the rebuttal deadline approaches, we remain committed to addressing any remaining questions you may have. We welcome further comments or suggestions for clarification and are ready to perform additional experiments if necessary. Your guidance will be invaluable in helping us further improve our work.

Best regards,

The Authors

---

### Note · Authors · 2025-08-12

# Dear ACs and Reviewers

We thank all reviewers for their constructive feedback and address each question below with a summary of reviews and rebuttals.

---

## 📌 Paper in One Line
**IndustryEQA**: First EQA benchmark for safety-critical industrial warehouses in Isaac Sim with **1,344 QA pairs**, evaluating factual accuracy and reasoning.

---

## ✅ Positives
- **Novel Domain:** First industrial EQA benchmark beyond household settings.
- **Safety Focus:** ~50% safety-related QAs, OSHA-inspired hazards, dynamic agents.
- **High-Quality Data:** Vivid scenes, LLM + human-refined QA generation.
- **Comprehensive Evaluation:** 15+ VLLMs, dual metrics, extensive ablations.
- **Relevance:** Real-world industrial applicability, clear differentiation.

---

## ⚠️ Negatives & 💡 Rebuttals

1. **Simulation Gap:** Fully simulated hazards.
   → Hazards modeled from OSHA guidelines & real warehouse photos; simulation chosen for safety, cost, and privacy. Asset library will be open-sourced.

2. **No Active Track:** Episodic only.
   → Chosen to control perception/reasoning evaluation; active track is a high-priority future extension.

3. **QA Bias:** OSHA + LLM bias risk.
   → Distribution mainly shaped by OSHA hazard design & available Isaac Sim assets; framework adaptable to other safety regulations.

4. **No Industrial SOTA Comparison:**
   → QA-TOOLBOX is small-object assembly–focused; not comparable. Closed-source VLLMs here outperform all open-source models.

5. **No Human Baseline:**
   → Human study (100 samples) shows higher Direct Scores than VLLMs; slightly lower Reasoning Scores due to concise answer styles.

6. **No Scene Density Analysis:**
   → Correlation analysis found no clear link; complexity of mixed object types noted.

7. **Scoring Reliability:**
   → ICC 0.837–0.953 shows high inter-LLM agreement.

8. **Placement Methodology:**
   → Real-photo/OSHA-inspired scenes; realistic worker tasks.

9. **Safety Definition:**
   → OSHA hazards; reasoning about *potential* risks vs anomaly detection.

10. **Reasoning Score Process:**
    → LLM prompts assess facts, logic, reasoning; all Q&A pairs manually verified to minimize ambiguity.

11. **Gemini 2.5 Pro Omission:**
    → Initially excluded to avoid unfair advantage; now included, achieving SOTA performance

---

## 📂 Additional Commitments
- Release 3D warehouse scenarios.
- Highlight unique industrial QA challenges.
- Explore on-device efficiency in future work.

---

### Decision · Program_Chairs · 2025-09-18

**Decision:**

Accept (poster)

**Comment:**

All reviewers incline to accept the submission thanks to the novelty, comprehensive baseline and evaluation suite, and clear presentation. The AC agrees to the reviewers and recommends to accept the submission.